# Beyond Chunks and Graphs: Retrieval-Augmented Generation through Triplet-Driven Thinking

## Abstract

Retrieval-augmented generation (RAG) is critical for reducing hallucinations and incorporating external knowledge into Large Language Models (LLMs). However, advanced RAG systems face a trade-off between performance and efficiency. Multi-round RAG approaches achieve strong reasoning but incur excessive LLM calls and token costs, while Graph RAG methods suffer from computationally expensive, error-prone graph construction and retrieval redundancy. To address these challenges, we propose $T^2RAG$, a novel framework that operates on a simple, graph-free knowledge base of atomic triplets. $T^2RAG$ leverages an LLM to decompose questions into searchable triplets with placeholders, which it then iteratively resolves by retrieving evidence from the triplet database. Empirical results show that $T^2RAG$ significantly outperforms state-of-the-art multi-round and Graph RAG methods, achieving an average performance gain of up to 11% across six datasets while reducing retrieval costs by up to 45%. Our code is available at `https://anonymous.4open.science/r/T2RAG-DF75`.

## 1 Introduction

Large Language Models (LLMs) have become central to open-domain question answering (QA) systems, owing to their vast stores of parametric knowledge and remarkable instruction-following capabilities (Yue, 2025; Gu et al., 2024b). However, their effectiveness is often undermined by critical challenges such as catastrophic forgetting and hallucination, particularly when addressing questions that require access to evolving, real-world knowledge (Gu et al., 2024a; Huang et al., 2025; Zhong et al., 2023). Consequently, Retrieval-Augmented Generation (RAG) has emerged as a robust paradigm to mitigate these issues (Lewis et al., 2020; Gao et al., 2023) by retrieving relevant documents from an external knowledge corpus.

However, standard RAG systems, which rank document chunks by query similarity (Karpukhin et al., 2020; Sawarkar et al., 2024; Khattab & Zaharia, 2020), are effective for simple questions but fail on complex ones that require multi-hop reasoning (Tang & Yang, 2024). This failure occurs because queries often lack the necessary intermediate entities to connect information across different chunks (Shen et al., 2024), and important details can be lost in the *compression loss* of long chunk embeddings (Zhang et al., 2024b).

To address these issues, two primary research directions have emerged, each with its own challenges. **Multi-Round RAG** leverages the LLM's reasoning abilities by decomposing complex questions into sequential sub-queries. While effective at traversing multi-hop knowledge paths, it is time and *token-consuming*, often requiring numerous (3-6) LLM calls in each round (Trivedi et al., 2023; Xu et al., 2025; Shen et al., 2024), and up to around 8 rounds in total (Trivedi et al., 2023). Additionally, it also faces the challenge of *compression loss*. On the other hand, **Graph RAG** (Edge et al., 2024; Han et al., 2024; Peng et al., 2024) structures the corpus into a knowledge graph to retrieve logically connected information. However, this approach is hindered by an expensive and error-prone graph construction process due to *entity ambiguity* issue (Hoffart et al., 2014), *redundancy* in retrieval from high-degree nodes (Peng et al., 2024), and the difficulty LLMs face when understanding the graph structures (Chai et al., 2023). The challenges of the above two lines of work lead to a question:

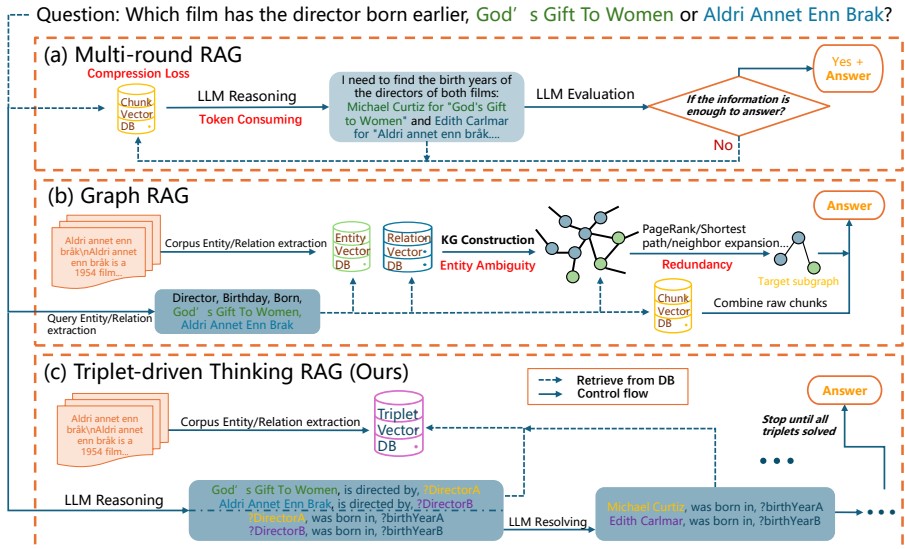

Figure 1: A comparison of three RAG paradigms, with their primary challenges highlighted in red. **(a) Multi-round RAG** employs an iterative loop to retrieve large text chunks, but is hampered by *compression loss* from vector embeddings and *high token consumption* during reasoning. **(b) Graph RAG** constructs a knowledge graph to retrieve answers, but is vulnerable to *entity ambiguity* during creation and *retrieval redundancy* from high-degree nodes. **(c) T$^2$RAG** decomposes a query into triplets with "?" placeholders and iteratively resolves them by retrieving evidence from a triplet database (DB) until all of them are resolved.

*Can we directly use the triplets as the fundamental unit of RAG, thus avoid the entity' ambiguity caused by entity level and compression loss caused by chunks level?*

Motivated by this question, we propose T$^2$RAG (Triplet-driven Thinking for Retrieval-Augmented Generation), a novel framework that fundamentally re-architects the RAG pipeline and moves beyond traditional chunk-based or graph-based retrieval by operating directly on atomic knowledge triplets. *Unlike Graph RAG*, it completely sidesteps the costly, time-consuming, and error-prone process of offline knowledge graph construction. Instead of building an explicit graph, T$^2$RAG operates on a graph-free knowledge base of atomic propositions, thus avoiding the high indexing costs and potential for retrieval errors caused by inaccurate graph links. *Simultaneously*, it tackles the excessive token consumption and latency that plagues Multi-round RAG systems. Rather than generating verbose, natural language reasoning chains at each step, T$^2$RAG leverages the LLM to think in a more structured, efficient manner. It expands complex questions into "searchable triplets" containing specific placeholders for unknown entities. The system then iteratively retrieves context to resolve these triplets. This design maintains a lean, structured state transition between iterations, passing only compact triplets instead of verbose text. This triplet-centric design ensures a tight coupling between retrieval and reasoning, retaining powerful multi-hop capabilities while dramatically reducing token overhead and enhancing performance. Our main contributions are as follows:

- We introduce a novel RAG framework that directly leverages triplets as the fundamental unit for indexing, retrieval, and reasoning, moving beyond the limitations of chunk-based and explicit graph-based approaches.
- We demonstrate that our method achieves state-of-the-art performance on various types of QA benchmarks, outperforming leading models in both the Multi-Round RAG and Graph RAG.
- We also significantly improve the efficiency. Our method reduces inference time and token consumption by up to 45% compared to other multi-round methods and even achieves an efficiency comparable to that of single-round approaches.

## 2 PRELIMINARIES

The task of open-domain question answering (ODQA) was formally introduced in the 1999 Text REtrieval Conference (TREC) QA track (Voorhees & Tice, 2000). Initially, it was defined as a factoid

QA task: Given a large corpus of unstructured documents, the goal was to extract a small text snippet containing the correct answer to a factual question. While the scope of ODQA has since expanded to include summarization and open-ended (Reja et al., 2003) tasks (Edge et al., 2024; Xiang et al., 2025), factoid QA remains a significant challenge, evidenced by poor performance (below 50%) on complex, multi-hop datasets like MusiQue (Trivedi et al., 2023). Consequently, this paper focuses on advancing the state-of-the-art in factoid QA.

**Factoid QA Task.** Assume our collection contains $D$ documents $d_1, d_2, \ldots, d_D$. We split each document into passages of equal token length or applying expert split if it exists, yielding $M$ total chunks $\mathcal{C} = \{c_1, c_2, \ldots, c_M\}$, where each chunk $c_i$ can be viewed as a token sequence $(w_1^{(i)}, w_2^{(i)}, \ldots, w_{|c_i|}^{(i)})$. Given a question $q$, the goal is to find a combination of tokens $(w_{c_m}^{(j)}, \ldots, w_{c_{m+k}}^{(j)})$ drawn from multiple chunks that collectively contain the information necessary to answer $q$ while minimizing irrelevant noise to avoid hallucination. The answer must be **exact one entity** in our setting, such as persons, organizations, or locations or yes/no. Typically, a retriever $R : (q, \mathcal{C}) \to \mathcal{C}_F$ is a function that takes a question $q$ and the corpus $\mathcal{C}$ as input and returns a much smaller set of chunks $\mathcal{C}_F \subset \mathcal{C}$, where $|\mathcal{C}_F| = k \ll |\mathcal{C}|$. For a fixed $k$, a retriever can be evaluated in isolation using top-$k$ retrieval accuracy with respect to labeled golden chunks.

**Retrieval Granularity.** The preceding formulation assumes the retrieval unit is the chunk, which is a common setting (Karpukhin et al., 2020). However, recent works especially Guo et al. (2024); Fan et al. (2025) argue that chunks often contain a mix of relevant and irrelevant details, and a finer granularity is needed for complex queries (Zhang et al., 2024b). Inspired by work in Knowledge Graphs (KGs) (Ji et al., 2021), the fundamental unit of retrieval can be refined to more atomic elements:

- **Entities** $(e_1^{(i)}, e_2^{(i)}, \ldots, e_{|c_i|}^{(i)})$: Named entities such as persons, organizations, or locations.
- **Triplets** $(t_1^{(i)}, t_2^{(i)}, \ldots, t_{|c_i|}^{(i)})$: Structured facts represented as a (subject,predicate,object) tuple.
- **Propositions** $(p_1^{(i)}, p_2^{(i)}, \ldots, p_{|c_i|}^{(i)})$: Atomic statements or facts, often by converting triplets into natural language sentences.

Propositions, which encapsulate a complete fact in a single sentence, are often considered to have greater semantic utility for modern embedding models compared to isolated entities or structured triplets (Zhang et al., 2024b). Our work explores leveraging this fine-grained units for improved retrieval and reasoning.

# 3 RELATED WORK

We group recent RAG efforts into *multi-round*, and *graph-enhanced* RAG, each adding more interaction or structured reasoning and paving the way for the fine-grained design of T²RAG.

**Multi-round RAG.** Due to missing intermediate entities problem we mentioned in Section 1 more and more works follow a multi-round paradigm, which enables the LLMs infer the intermediate information thus better retrieve the final answer. Some works focus on the query side. Khot et al. (2023) decompose multi-hop questions into single-hop sub-queries that are solved sequentially. Yao et al. (2023) propose ReAct, interleaving chain-of-thought (CoT) (Wei et al., 2022) steps with search actions issued by the LLM. Similarly, Query2Doc (Wang et al., 2023b) expanding queries into concise triplets to cut token usage while preserving recall. Another line of works relies on the generated intermediate results for next iteration. Beam Retrieval (Zhang et al., 2024a) jointly training an encoder and classifiers to keep multiple passage hypotheses across hops. FLARE (Jiang et al., 2023) forecasts upcoming sentences to decide when fresh retrieval is needed during long-form generation. IRCoT (Trivedi et al., 2023) and ITER-RETGEN (Shao et al., 2023), alternately expanding a CoT and fetching new evidence to answer multi-step questions. Adaptive QA (Xie et al., 2023) create an adaptive framework that picks the simplest effective retrieval strategy according to query complexity. *Despite these advances, few efforts explicitly aim to reduce token costs or number of LLM calls during multi-round RAG. Previous methods expand query or generates CoT with long sentences in each round. In contrast, our work minimizes token consumption by formulating query expansions as triplets and simplifying reasoning steps as triplets resolving.*

**Graph RAG.** @revised: A significant line of research approaches ODQA by structuring knowledge into graphs, subsequently adopting methods from Knowledge Graph QA (KGQA) (Ma et al., 2025). Early KGQA methods primarily focused on decomposing queries or performing multi-round, LLM-evaluated traversals starting from seed nodes (Luo et al., 2024; Sun et al., 2024; Cheng et al., 2024; Mavromatis & Karypis, 2022). Other works rely on training specialized retrievers (Li et al., 2025), GNN models (Mavromatis & Karypis, 2024), or LLMs (Tian et al., 2024) to achieve accurate retrieval. However, ODQA is distinct from KGQA in two key aspects: 1) it is fundamentally grounded in unstructured text, and 2) it prioritizes final answering accuracy over pure retrieval precision, as it allows to use knowledge inherent in the LLM.@ Semi-Structured CoT addresses ODQA by combining IRCoT and KGQA methods to generate reasoning chains that feature a single, fixed placeholder resolved by KG techniques (Su et al., 2024). The application of this paradigm to general ODQA was popularized by systems named GraphRAG (Edge et al., 2024) that construct a knowledge graph entirely with LLMs and use community detection for hierarchical summarization and retrieval. Subsequent work has aimed to make this process more efficient. For instance, LightRAG (Guo et al., 2024) introduces a dual-level retrieval system combining graph structures with vector search to improve knowledge discovery. Targeting resource-constrained scenarios, MiniRAG (Fan et al., 2025) builds a heterogeneous graph of text chunks and named entities, enabling lightweight retrieval suitable for Small Language Models. To tackle the common challenge of entity merging, HippoRAG (Gutiérrez et al., 2025a) and HippoRAG2 (Gutiérrez et al., 2025b) create synonym links between similar entity nodes and employs a PageRank (Haveliwala, 1999) algorithm for final node selection. *Despite these advances, a central challenge for Graph RAG remains the costly and error-prone nature of graph construction from unstructured text.*

Our method, $T^2$RAG, skips the costly and error-prone graph construction required by Graph RAG while retains the multi-hop reasoning power by Multi-round RAG. It also dramatically reduces token overhead by constraining both query expansion and intermediate generation. Besides, some works in ODQA such as GEAR (Shen et al., 2024) also employ a triplet search component. These methods typically rely on neighbor expansion, which involves retrieving all other triplets that share a head or tail entity. A key drawback of this approach is that accurately identifying and linking the same entity across different contexts is often inaccurate and computationally expensive.

## 4 METHODOLOGY

### 4.1 OVERVIEW

Our proposed method, **$T^2$RAG** (**T**riplet-driven **T**hinking **RAG**), is a novel paradigm for resolving complex, multi-hop, factoid QA tasks. Unlike conventional RAG systems that operate on coarser document chunks or complex graph structures, **$T^2$RAG** is designed to operate directly on atomic knowledge propositions derived from triplets, fostering an intrinsic alignment between knowledge representation and LLM reasoning. This framework operates in two stages: an offline indexing focused on systematic knowledge distillation, and an online retrieval characterized by iterative, adaptive triplet resolution. This principled design ensures both fine-grained retrieval for accuracy and a lean, efficient reasoning process.

### 4.2 OFFLINE INDEXING: CONSTRUCTING A GRAPH-FREE KNOWLEDGE BASE

The goal of the offline stage is to transform a raw text corpus $\mathcal{C}$ into a efficiently searchable knowledge base of atomic propositions. The motivation for adopting **proposition level** granularity is two fold: 1) Compared to the entity level, each proposition encodes an entire, unambiguous fact. 2) Compared to the chunk level, it also avoids the compression loss hindering the retrieval of details.

**Canonical Triplet Generation.** For each document chunk $c_i \in \mathcal{C}$, we employ an information extraction model, $LLM_{IE}(\cdot)$, to identify key facts. This model performs Open Information Extraction (OpenIE) (Martinez-Rodriguez et al., 2018) to extract a set of knowledge triplets $\mathcal{T}_i = \{t_1^{(i)}, t_2^{(i)}, \dots\}$. Each triplet $t_j^{(i)}$ is formalized as a canonical knowledge triplet $(subject, predicate, object)$ that represents a single factual statement. All extracted triplets are then aggregated into a global set for the entire corpus $\mathcal{T}_{total} = \bigcup_{i=1}^{M} \mathcal{T}_i$, where $M$ is the total number of extracted triplets. *To demonstrate the power of using triplets as a foundational unit, we employ an off-the-shelf triplet generation method.*

*While developing a more accurate and comprehensive extraction technique is outside the scope of this work, we provide a detailed error analysis of the method used in the Appendix B.8*

**Triplet Embedding.** To render these canonical triplets semantically actionable for dense retrieval, we are inspired by verbalization techniques (Oguz et al., 2020; Baek et al., 2023) to convert each triplet $t \in \mathcal{T}_{total}$ into a natural language sentence, termed a *proposition p*, simply by concatenating its components (e.g., "subject predicate object"). This seemingly straightforward verbalization is a deliberate design choice: it maximizes the semantic utility for embedding models, facilitating effective and contextually rich retrieval compared to isolated entities.

**Triplet Vector DB Construction.** The resulting flat list of propositions $\mathcal{P}_{total} = \{p_1, p_2, \ldots, p_M\}$ is then encoded into dense vector representations using a high-performance embedding model $E(\cdot)$. For efficient real-time access, these vectors can be subsequently indexed using a highly optimized vector search library (FAISS) (Douze et al., 2024), creating an index $\mathcal{I}$ that enables rapid similarity search across all propositions in the corpus. This vector DB is still called **Triplet Vector DB** as it keeps original text of triplets. We also save the mapping from those propositions to their source chunks because the original text is proved necessary in most of Graph RAG works (Guo et al., 2024; Fan et al., 2025). This pre-computation creates a fine-grained, semantically enriched knowledge index **without the overhead of explicit graph structures**.

The constructed proposition index, while offering significant advantages in terms of cost and construction fidelity, introduces a critical challenge: *how to effectively navigate complex, multi-hop questions that typically rely on graph traversals?* In the subsequent subsection, we introduce our novel online retrieval stage, where the LLM's triplet-driven thinking and adaptive iterative resolution strategically compensate for the graph traversals and the path-based reasoning.

### 4.3 ONLINE RETRIEVAL: ITERATIVE TRIPLETS RESOLUTION

The online retrieval stage is an iterative process that dynamically builds the context containing both the triplets and chunks needed to answer user queries. The overall retrieval process is shown in Figure 2.

**Step 1: Structured Query Decomposition.** Given an initial query $q$, we first use an LLM to perform a structured decomposition where the LLM identifies the specific, atomic knowledge Triplets (denoted as $\mathcal{T}_q$) that must be answered to address the overall query. Critically, these derived triplets contain explicit placeholders ('?') for unknown entities. Based on the precise number of these placeholders, we categorize these initial triplets into three types:

- **Resolved Triplets** ($\mathcal{T}_{\text{resolved}}$): Triplets with **zero** placeholders, representing fully known facts that require no further search.
- **Searchable Triplets** ($\mathcal{T}_{\text{searchable}}$): Triplets with exactly **one** placeholder. This specificity, with two known elements, facilitates focused and accurate searches.
- **Fuzzy Triplets** ($\mathcal{T}_{\text{fuzzy}}$): Triplets with two or more placeholders. These are inherently too ambiguous for search with the at most one element. It requires resolution in subsequent iterations to upgrade to **searchable** or **resolved**.

This explicit categorization ensures that later retrieval efforts are always focused and efficient.

**Step 2: Multi-Round Triplet Resolution with Triplet Retrieval.** In this step, we will resolve the query triplets, i.e., try to eliminate all "?" placeholders step by step by RAG. Considering different complexity of queries and their triplets, we adopt an adaptive retrieval strategy instead of a fixed top-$k$. We also observed most of multi-hop questions cannot be specifically retrieved by the query itself as illustrated in Figure 1, which necessitate the multi-round paradigm.

**Step 2.1: Triplet-Based Adaptive Retrieval.** The current set of searchable triplets $\mathcal{T}_{\text{searchable}}^{(l)}$ are first converted into *query propositions* by simply concatenating the elements without the placeholder. These propositions are then embedded, using the same embedding model

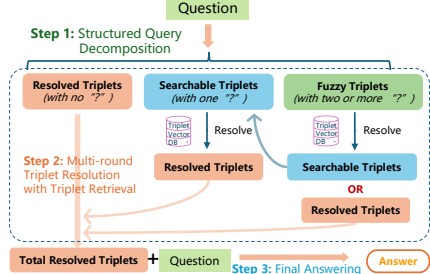

Figure 2: Online retrieval stage.

$E(\cdot)$ in the indexing stage, and used to query the proposition index $\mathcal{I}$. Unlike prior methods that retrieve a fixed top-$k$ of propositions or triplets (Baek et al., 2023; Guo et al., 2024), our retrieval process is critically adaptive in two synergistic ways to ensure both relevance and informational diversity: **First**, our method retrieves with the triplets while constrain the process by chunks. More specifically, the retrieval dynamically continues until context from $k$ unique source chunks of triplets has been retrieved. **Second**, we aggregate retrieval candidates from all query propositions into a unified pool, ranking them globally by similarity scores, rather than allocating separate budgets to each proposition. These adaptive strategies ensure robustness to varying query complexity, allowing difficult questions to naturally draw from a wider range of propositions. Finally, the retrieval process returns the set of retrieved propositions $\mathcal{P}_{\text{retrieved}}^{(l)}$ and their corresponding source chunks $\mathcal{C}_{\text{retrieved}}^{(l)}$. The necessity of reading original chunks to complete details missing from triplets is widely acknowledged in the field (Fan et al., 2025; Guo et al., 2024).

**Step 2.2: Resolving Triplets with Retrieved Context.** This step leverages the retrieved content to advance the query's resolution. We prompt the LLM to populate the placeholders within these triplets using the provided context. The retrieved propositions ($\mathcal{P}_{\text{retrieved}}^{(l)}$) and and their source chunks ($\mathcal{C}_{\text{retrieved}}^{(l)}$) serve as context for an LLM call. This is designed to either upgrade a searchable triplet to a fully resolved one by filling in its single placeholder, or to transform a fuzzy triplet into a searchable or directly to a resolved one by filling in one or more of its multiple placeholders. This resolution process reduces the ambiguity of existing triplets and makes it suitable for subsequent targeted retrieval. The process is shown in Figure 2 and a detailed example is in Appendix E.

@revised: **Step 2.3: State Update.** Following the triplet resolution step, the system updates its internal state for the subsequent iteration. The collection of **resolved triplets** is monotonically augmented with any facts verified in the current round. Crucially, to maintain a focused retrieval scope, the system targets only the **newly searchable triplets** for the next iteration, rather than re-scanning previous queries. Simultaneously, the queue of **fuzzy triplets** is updated by pruning any items that were either successfully resolved or promoted to a searchable state in the current round. In cases where no searchable triplets are identified, the system employs a **fallback mechanism** to ensure robustness. It utilizes the embedding of the current natural language query to perform dense retrieval directly against the triplet vector database.@

This highly structured transition is central to our method's efficiency. By passing compact triplets between iterations—rather than the verbose Chain-of-Thought reasoning used by approaches like IRCoT—we dramatically reduce token overhead. Furthermore, this design creates a powerful synergy: the LLM generates reasoning gaps in the same format (triplets) as the retrieval index, ensuring strong semantic alignment between the resolution and retrieval stages.

**Step 2.4: Termination and Final Answer Synthesis.** @revised: The iterative loop continues until one of three conditions is met: (1) the queues for both searchable and fuzzy triplets are empty, indicating the reasoning chain is complete; (2) no newly searchable triplets are generated and no remaining fuzzy triplets, triggering an early stop; or (3) a pre-defined maximum number of iterations ($N$) is reached. Upon termination, the system aggregates the verified facts to generate the final answer via one of two pathways:@ **(a) Successful Completion:** If the loop terminates naturally because all triplets were resolved, the LLM generates the answer conditioned on the original query and the set of fully resolved triplets. **(b) Forced Termination (Early Stop or Max Limit):** If the process halts due to the iteration limit or a failure to generate new searchable triplets, the system constructs the best possible context by combining all accumulated triplets (both resolved and remaining searchable ones).

@revised: By grounding the final generation primarily in verified, structured data rather than raw retrieved text chunks, this method minimizes token consumption and reduces the risk of hallucination.@

## 5 EXPERIMENTS

### 5.1 DATASETS

To ensure a comprehensive evaluation, we select representative datasets for three distinct Open-Domain Question Answering (ODQA) categories: Simple QA, Multi-hop QA, and Domain-specific

Table 1: Main performance comparison on various types of QA datasets, showing Exact Match / F1 scores $\times 100$. The best result in each column is in **bold**, and the second best is underlined. Bootstrap testing is used to assess significance with 5,000 times repeat. The dagger symbol ($\dagger$) indicates that our method significantly outperforms the best baseline (IRCoT under Gemini-2.5-flash and HippoRAG2 under GPT-4o-mini) with a p-value less than 0.05.

| | Simple QA | Multi-Hop QA | | | Domain-Specific QA | | Average | |
|---|---|---|---|---|---|---|---|---|
| **Method** | PopQA | 2Wiki | MuSiQue | HotpotQA | Story | Medical | EM | F1 |
| *Gemini-2.5-flash* | | | | | | | | |
| NOR | 32.4 / 35.7 | 48.1 / 55.6 | 16.3 / 26.5 | 40.5 / 52.3 | 10.3 / 17.1 | 23.1 / 46.0 | 28.4 | 38.9 |
| BM25 | 50.2 / 55.6 | 28.2 / 30.7 | 7.9 / 10.7 | 40.8 / 49.3 | 26.2 / 35.3 | 22.2 / 37.8 | 29.3 | 36.6 |
| Standard | 51.8 / 59.5 | 33.1 / 39.0 | 28.1 / 36.2 | 52.1 / 63.1 | 31.0 / 42.2 | 19.4 / 41.5 | 35.9 | 46.9 |
| HippoRAG2 | 52.1 / 60.1 | 44.3 / 51.2 | 29.1 / 38.3 | 52.1 / 64.1 | 33.1 / 44.1 | 27.8 / 58.2 | 39.8 | 52.7 |
| RAPTOR | 52.3 / 56.8 | 36.3 / 41.1 | 31.8 / 39.7 | 60.9 / 72.7 | 46.2 / 59.0 | 34.2 / 58.1 | 43.6 | 54.6 |
| IRCoT | 51.2 / 58.7 | 61.6 / 71.7 | **39.7** / **49.8** | 61.2 / **77.3** | 40.3 / 57.3 | 26.1 / 56.1 | 46.7 | 61.8 |
| T$^2$RAG | **56.6**$^\dagger$ / **62.4**$^\dagger$ | **69.3**$^\dagger$ / **77.5**$^\dagger$ | 39.1 / 49.1 | **62.3** / 73.2 | **46.7**$^\dagger$ / **59.5**$^\dagger$ | **36.0**$^\dagger$ / **61.4**$^\dagger$ | **51.7** | **63.9** |
| *GPT-4o-mini* | | | | | | | | |
| NOR | 28.7 / 31.4 | 28.0 / 34.1 | 10.2 / 20.3 | 28.8 / 38.6 | 11.5 / 18.9 | 19.3 / 44.2 | 21.1 | 31.3 |
| BM25 | 47.6 / 54.8 | 42.9 / 48.2 | 15.3 / 21.1 | 47.2 / 57.6 | 29.0 / 38.5 | 25.9 / 43.6 | 34.7 | 44.0 |
| Standard | 51.9 / 60.0 | 53.1 / 60.2 | 31.2 / 44.3 | 58.0 / 71.1 | 27.3 / **60.1** | 27.0 / 59.9 | 41.4 | 59.3 |
| HippoRAG2 | 52.2 / 60.2 | 59.6 / 69.3 | 34.1 / **48.1** | **58.1** / 71.1 | 41.2 / 58.3 | 28.1 / 59.4 | 45.6 | **61.1** |
| RAPTOR | 54.6 / 60.1 | 38.2 / 49.0 | 28.6 / 40.8 | 57.9 / **71.4** | **44.8** / 59.6 | **36.7** / **63.7** | 43.5 | 57.4 |
| IRCoT | 45.3 / 54.7 | 60.7 / 74.3 | 34.1 / 47.6 | 55.7 / 71.2 | 36.1 / 51.8 | 25.1 / 52.9 | 42.8 | 58.8 |
| T$^2$RAG | **55.8**$^\dagger$ / **63.2**$^\dagger$ | **66.7**$^\dagger$ / **74.4**$^\dagger$ | **34.3** / 45.6 | 54.2 / 67.3 | 38.7 / 50.1 | 33.5$^\dagger$ / 60.4 | **47.2** | 60.2 |

QA. For the first two categories, we follow the experimental setup from HippoRAG2 (Gutiérrez et al., 2025b). We use PopQA (Mallen et al., 2023) for simple questions. For multi-hop questions, we use 2Wiki-MultihopQA (2Wiki) (Ho et al., 2020), MuSiQue (Trivedi et al., 2022), and HotpotQA (Yang et al., 2018). For each of these datasets, we use the same sample of 1,000 questions as the prior work (Gutiérrez et al., 2025b). For domain-specific evaluation, we adapt two datasets from the GraphRAG-Bench (Xiang et al., 2025). We isolate the factoid questions from the two datasets, Story and Medical, and use an LLM to shorten the ground-truth answers, enabling more precise evaluation. Detailed statistics for all datasets are provided in Table 3.

## 5.2 BASELINES AND IMPLEMENTATION DETAILS

To evaluate our approach, we select three strong baselines representing state-of-the-art methods across major RAG categories. For Graph RAG, we choose **HippoRAG2** (Gutiérrez et al., 2025b) for its recognized efficiency and effectiveness. For summarization-based RAG, we use **Raptor** (Sarthi et al., 2024), a pioneering method that outperforms most Graph RAG approaches in recent benchmarks (Zhou et al., 2025). Lastly, for Multi-Round RAG, we include the prominent **IRCoT** (Trivedi et al., 2023) method. **NOR** method means the non-retrieval method that directly answers the question. **Standard** RAG retrieves chunks with an embedding model and uses them to generate an answer.

To ensure a fair comparison, all methods are configured with the same foundational models: NV-Embed-v2 (Lee et al., 2024) for embeddings and either Gemini-2.5-flash or GPT-4o-mini as the LLM for all offline indexing and online retrieval stages. For datasets lacking expert annotations, we employ a standard chunking strategy of 1200 tokens with a 100-token overlap. For the top-$k$ of chunk retrieval, we set $k = 5$ for all methods. For the multi-round methods (T$^2$RAG and IRCoT), we set a maximum of $N = 3$ iterations and keeps the $k = 5$ in each iteration. Following standard practices (Trivedi et al., 2023), we evaluate end-to-end QA performance using Exact Match (EM) and F1 scores. We focus specifically on these end-to-end QA metrics, as retrieval performance is difficult to compare directly when the number of retrieved passages is adaptive. Except for the performance comparisons, all results presented in the subsequent sections are obtained using GPT-4o-mini. Further experimental details are available in Appendix B.

## 5.3 RESULTS

We unfold our analysis of experimental results by answering Research Questions (RQ) below.

**RQ1: How does T$^2$RAG perform against baselines?** As shown Table 1, T$^2$RAG achieves state-of-the-art performance, stems from several key advantages. **First**, our method achieves state-of-the-art overall performance, leading in both average EM and F1 scores across the two LLM backbones,

except for the second place in F1 by GPT-4o-mini. Notably, its advantage in EM is particularly pronounced, a strength we attribute to the precision of our triplet-based retrieval, which excels at identifying the exact entities required for factoid QA. This adaptability is further demonstrated by its consistently strong results on domain-specific datasets, underscoring the universality of the underlying reasoning framework. **Second**, its superiority is most pronounced on Multi-hop QA datasets like 2Wiki. It not only surpasses all single-round baselines by a large margin but also outperforms the multi-round baseline, IRCoT, by over 7.7% and 5.4% in EM with Gemini-2.5-flash and GPT-4o-mini, respectively. This highlights the effectiveness of its triplet-driven mechanism for complex reasoning. **Finally**, the method demonstrates a powerful synergy with reasoning LLMs. Its performance is significantly higher when paired with Gemini-2.5-flash compared to GPT-4o-mini. As shown in Figure 8, we extend the experiments on 2 reasoning LLMs (Gemini-2.5-flash and Gemini-2.5-pro) and 2 non-reasoning LLMs (GPT-4Oø-mini and Qwen3-Next-Instruct (Yang et al., 2025)). We compare the averaged EM and F1 score on PopQA dataset. The results suggest that our method can uniquely leverage the advanced **reasoning capabilities** of such models through its step-by-step guidance. Conversely, certain methods such as HippoRAG2 exhibit a decrease in performance when employing reasoning LLMs. We hypothesize this occurs because relegating the LLM to a simple filtering task does not fully harness its reasoning capabilities. Results on more datasets can be found in Appendix B.3.

**RQ2: What is the impact of the triplet resolution module?** To validate the effectiveness of our core "triplet-driven thinking" design, we analyze the final performance based on whether a query's underlying triplets are fully resolved. Figure 4 reveals a significant performance delta between these two outcomes. Across all three datasets, there is a strong correlation between successful triplet resolution and high performance. For instance, on the 2Wiki dataset, the F1 score for unresolved questions drops to 53% from 76%, with a similar sharp decline observed in EM scores. This result confirms that resolving all triplets is the key to success. We also use LLM to do an error analysis of the incorrect answer. Results shows the missing retrieval leads the reason of error and the hallucination is as low as 2%. The details are in Appendix B.9.

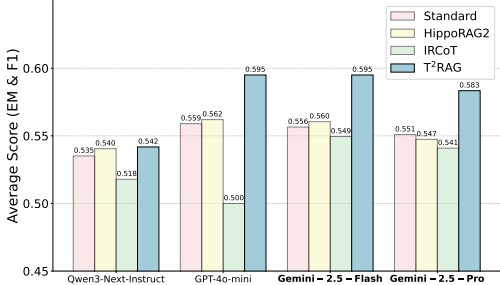

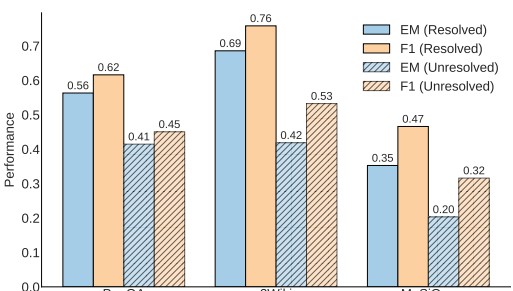

Figure 3: Performance on PopQA across different LLMs. Reasoning LLMs are in **bold**.

Figure 4: Performance vs. final resolution status across three datasets.

**RQ3: Which components of T$^2$RAG are important?** We conducted an ablation study to quantify the contribution of its two key components. The results in Table 5 reveal that both the iterative process and the use of chunks are important. The iterative reasoning module proves to be a critical component. Removing it (- single round) causes a significant performance degradation, particularly on multi-hop QA. For instance, F1 score on MuSiQue drops by a remarkable

Figure 5: Ablation results

| Method | PopQA | | 2Wiki | | MuSiQue | |
|---|---|---|---|---|---|---|
| | EM | F1 | EM | F1 | EM | F1 |
| T$^2$RAG | 56.0 | 63.0 | 66.0 | 74.0 | 33.0 | 45.0 |
| - single round | 54.8 ↓2.1% | 60.5 ↓4.0% | 51.0 ↓22.7% | 59.0 ↓20.3% | 15.0 ↓54.5% | 24.0 ↓46.7% |
| - w/o chunk | 41.1 ↓26.6% | 44.7 ↓29.0% | 62.0 ↓6.1% | 68.0 ↓8.1% | 21.6 ↓34.5% | 29.9 ↓33.6% |

54.5%. This demonstrates that the multi-round retrieval and resolution is essential for decomposing and solving complex problems. Similarly, removing the raw chunk text during the iteration, i.e, (- w/o chunk), is also substantially harms performance, confirming that the raw text complement missing details of triplets. This observation is aligned with Fan et al. (2025).

**RQ4: How does T$^2$RAG compare in terms of computational efficiency?** This analysis compares the computational cost of T$^2$RAG with baselines during both the one-time offline indexing and online

retrieval phases. To better visualize the online costs, the token and time values for the retrieval stage in Figure 6 are aggregated over 1,000 queries, assuming they are processed sequentially. Figure 6 illustrates a strategic trade-off. During **indexing stage**, T$^2$RAG's token consumption appears high because it processes the entire corpus into triplets. *However, this processing is merely the first step for many advanced Graph RAG methods (Edge et al., 2024; Guo et al., 2024; Fan et al., 2025). Their subsequent graph construction steps are far more costly.* For example, LightRAG and GraphRAG require around 6× and 10× the token consumption of the initial triplet extraction phase, respectively (Gutiérrez et al., 2025b). T$^2$RAG's indexing overhead remains highly competitive within this category. At the **retrieval stage**, T$^2$RAG is remarkably more efficient in both tokens and latency than the multi-round baseline, IRCoT. More notably, its efficiency is even comparable to single-round methods. This is because HippoRAG2 also invokes multiple LLM calls for filtering, while Raptor retrieves longer summaries than chunks. T$^2$RAG's efficiency stems from its design, which focuses on targeted search for triplets rather than processing large, noisy text chunks. In summary, T$^2$RAG achieves an acceptable indexing cost to deliver a highly efficient online system.

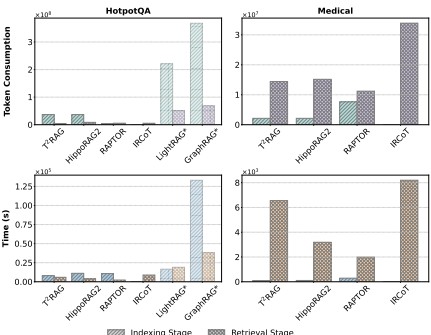 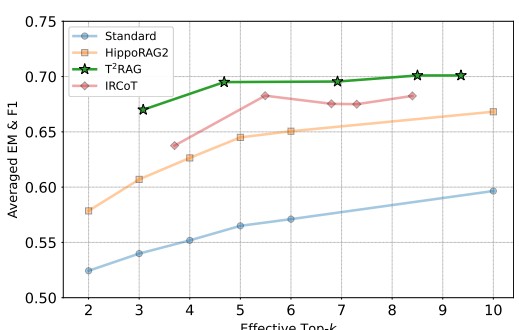

Figure 6: Comparison of token consumption and time. Token consumption equals to (input + 4×output). Results of LightRAG and GraphRAG are from Zhou et al. (2025).

Figure 7: Performance on 2Wiki vs. top-$k$. Multi-round methods are calibrated by $k×$ average number of iterations.

**RQ5: How does performance scale with the amount of retrieved context?** To investigate how T$^2$RAG's performance scales with context size, we compare it against other multi-round methods while varying the number of retrieved documents (top-$k$). Traditional RAG methods often rely on retrieving more context to find the correct answer, which can be inefficient. The trend in Figure 7 shows T$^2$RAG's performance is consistently high and robust to the value of top-$k$. It achieves the plateau (0.7) faster than other methods. In contrast, baselines like IRCoT and HippoRAG2 exhibit a strong dependence on a larger context window. This observation demonstrates its effectiveness does not rely on scaling up the volume of retrieved text but a more precise and specific triplet-based retrieval.

## 6 CONCLUSION

In this work, we proposed the Triplet-driven Thinking RAG (T$^2$RAG), a novel framework that embeds reasoning directly into the retrieval process. By decomposing complex queries into atomic triplets and resolving them step-by-step against a triplet knowledge base, our method consistently outperforms more complexly designed RAG systems. Our extensive experiments demonstrate that T$^2$RAG establishes a new state-of-the-art in factoid QA tasks, particularly on challenging multi-hop QA. This superior performance is achieved with remarkable online efficiency; the retrieval stage has significantly lower time and token consumption compared to other multi-round methods and maintains a comparable overhead to even single-round approaches. Furthermore, our results reveal a powerful synergy between T$^2$RAG's structured thinking process and the capabilities of advanced reasoning LLMs, highlighting a new path to unlock their full potential in this area. Looking forward, T$^2$RAG paves the way for more accurate and efficient RAG systems by shifting the paradigm from retrieving and generating unstructured contexts towards a more deliberate, reasoning-driven synthesis of atomic facts.

## STATEMENTS

### ETHICS STATEMENT

Our work focuses on developing a high-efficiency method for Retrieval-Augmented Generation (RAG) systems. The primary ethical benefit of this research is the potential reduction in energy consumption associated with these models, contributing to more sustainable computational practices. Furthermore, by increasing the speed at which accurate results are delivered, our method aims to improve user access to information. Our experimental methodology is designed to be robust and comprehensive, utilizing six datasets from diverse domains to ensure the generalizability of our findings. We verify the statistical significance of our results through p-value testing. To mitigate the risk of generating harmful or discriminatory content, the large language models (LLMs) used in our system have content filters in place. We are committed to transparency; full implementation details are provided in Appendix B.1. Our paper includes an extensive review of related literature to properly acknowledge and build upon previous work. The datasets used are all open-source and derived from public knowledge bases like Wikipedia, minimizing concerns related to privacy and confidential data.

### REPRODUCIBILITY STATEMENT

Our full implementation, along with instructions to reproduce all experimental results, is available at an anonymized code repository: `https://anonymous.4open.science/r/T2RAG-DF75`.

The datasets used in our experiments are sourced from public repositories. Specifically, the 2Wiki-MultihopQA, MusiQue, and HotpotQA datasets were obtained from the HippoRAG repository[1]. The PopQA dataset was downloaded from Hugging Face[2] and reformatted using a custom Python script to align with the other datasets. From each of these five datasets, we randomly sampled 1000 questions to construct our evaluation benchmark.

The Story and Medical datasets were sourced from the GraphRAG-Bench repository[3]. It is noted that the Story dataset is referred to as Novel in the source repository. For these, we selected only the "Fact Retrieval" level questions. To align the data with our factoid question-answering task, we employed an LLM to simplify the ground-truth answers using the following prompt:

> **Instruction:** You are an assistant for data processing. Your task is to simplify a given answer into a direct, concise response to the question. Follow these rules:
>
> 1. For yes/no questions, the simplified answer must be only "Yes" or "No", without any explanation.
>
> 2. For "wh-" questions (who, what, where, etc.), the simplified answer must be the specific entity or value requested.
>
> [Input] "question": "Does older age influence the risk of basal cell carcinoma?", "answer": "Older age is associated with higher risk of BCC."
>
> [Output] "answer": "Yes"

### USE OF LLMS

During the preparation of this manuscript, we utilized a large language model (LLM) to assist with improving grammar, clarity, and overall readability. The authors reviewed, edited, and take full responsibility for the final content, including the accuracy of all technical claims and citations.

---

[1] `https://github.com/OSU-NLP-Group/HippoRAG/tree/main/reproduce/dataset`

[2] `https://huggingface.co/datasets/akariasai/PopQA`

[3] `https://huggingface.co/datasets/GraphRAG-Bench/GraphRAG-Bench`

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

---

**Algorithm 1** T$^2$RAG: Online Iterative Triplet Resolution (Main Process)

---

**Input:** Query $q$, Triplet DB Index $\mathcal{I}$, LLM, Max Iterations $K$, Target unique chunks $k$, Triplet-to-Chunk-Map $\mathcal{M}_{\text{chunk}}$
**Output:** Final answer $a$

1:                                   ▷ Step 1: Structured Query Decomposition
2: $\mathcal{T}_{\text{resolved}}, \mathcal{T}_{\text{searchable}}, \mathcal{T}_{\text{fuzzy}} \leftarrow \text{LLM}_{\text{Decompose}}(q)$

3:                                   ▷ Step 2: Multi-Round Triplet Resolving Loop
4: **for** $l = 1 \rightarrow K$ **do**
5:     **if** $|\mathcal{T}_{\text{searchable}} \cup \mathcal{T}_{\text{fuzzy}}| = 0$ **then**
6:        **break**
7:     **end if**

8:                          ▷ Step 2.1: Call the Adaptive Retrieval (see Algorithm 2)
9:     $\mathcal{P}_{\text{retrieved}}, \mathcal{C}_{\text{retrieved}} \leftarrow \text{ADAPTIVERETRIEVE}(\mathcal{T}_{\text{searchable}}, \mathcal{I}, k, \mathcal{M}_{\text{chunk}})$

10:                          ▷ Step 2.2: LLM-based Triplets Resolution
11:     $\mathcal{T}_{\text{resolved}}^{(\text{new})}, \mathcal{T}_{\text{searchable}}^{(\text{new})} \leftarrow \text{LLM}_{\text{Resolve}}(\mathcal{T}_{\text{searchable}}, \mathcal{T}_{\text{fuzzy}}, \mathcal{P}_{\text{retrieved}}, \mathcal{C}_{\text{retrieved}})$

12:                                         ▷ Step 2.3: State Update
13:     $\mathcal{T}_{\text{resolved}} \leftarrow \mathcal{T}_{\text{resolved}} \cup \mathcal{T}_{\text{resolved}}^{(\text{new})}; \mathcal{T}_{\text{searchable}} \leftarrow \mathcal{T}_{\text{searchable}}^{(\text{new})}; \mathcal{T}_{\text{fuzzy}} \leftarrow \mathcal{T}_{\text{fuzzy}} \setminus (\mathcal{T}_{\text{resolved}}^{(\text{new})} \cup \mathcal{T}_{\text{searchable}}^{(\text{new})})$
14: **end for**

15:                                            ▷ Step 3: Final Answering
16: **if** $|\mathcal{T}_{\text{searchable}} \cup \mathcal{T}_{\text{fuzzy}}| = 0$ **then**
17:     $\mathcal{T}_{\text{context}} \leftarrow \mathcal{T}_{\text{resolved}}$
18: **else**
19:     $\mathcal{T}_{\text{context}} \leftarrow \mathcal{T}_{\text{resolved}} \cup \mathcal{T}_{\text{searchable}}$
20: **end if**
21: $a \leftarrow \text{LLM}_{\text{Answer}}(q, \mathcal{T}_{\text{context}})$
22: **return** $a$

---

# A METHODOLOGY

As the T$^2$RAG consists of several steps with clear control flow, we illustrate it by the following pseudo algorithm.

# B EXPERIMENTS

## B.1 DETAILED IMPLEMENTATIONS

For all experiments, we set the Large Language Model (LLM) temperature to 0 to ensure deterministic and reproducible outputs. Local embedding generation was performed on a single NVIDIA L40S GPU.

A key aspect of our benchmark is the standardization of the final answer format. We modified the prompt for all methods to include a specific format template, which yielded a significant performance boost compared to baseline implementations in other studies (Gutiérrez et al., 2025a; Xiang et al., 2025). In those works, methods such as RAPTOR and IRCOT consistently performed about 10% lower than graph-based RAG approaches. Furthermore, in our implementation of the **RAPTOR**, we replaced the original Gaussian Mixture Model (GMM) for clustering with K-Means. This decision was based on the superior computational efficiency of K-Means, which has been demonstrated to produce results of similar quality for this type of task (Zhou et al., 2025). The cluster size is set to 10 and level is set to 3 following the benchmark (Zhou et al., 2025). Our implementation of **IRCoT**

---

**Algorithm 2** Adaptive Triplet Retrieval

---

**Require:** Searchable triplets $\mathcal{T}_{\text{searchable}}$, Index $\mathcal{I}$, Target chunks $k$, Map $\mathcal{M}_{\text{chunk}}$
**Ensure:** Retrieved propositions $\mathcal{P}_{\text{retrieved}}$, Retrieved chunks $\mathcal{C}_{\text{retrieved}}$

---

1: **function** ADAPTIVERETRIEVE($\mathcal{T}_{\text{searchable}}, \mathcal{I}, k, \mathcal{M}_{\text{chunk}}$)
2:     $P_{\text{candidates}} \leftarrow \emptyset$
3:     **for** $t \in \mathcal{T}_{\text{searchable}}$ **do**
4:         query_prop $\leftarrow$ Concatenate($t$)
5:         query_vec $\leftarrow E$(query_prop)
6:         $P_{\text{candidates}} \leftarrow P_{\text{candidates}} \cup \text{Search}(\mathcal{I}, \text{query\_vec}, N)$
7:     **end for**
8:     Sort $P_{\text{candidates}}$ globally by similarity score

9:     $\mathcal{P}_{\text{retrieved}} \leftarrow \emptyset$;   unique_chunk_ids $\leftarrow \emptyset$
10:     **for** $p \in$ sorted $P_{\text{candidates}}$ **do**
11:         **if** |unique_chunk_ids| $\geq k_{\text{chunks}}$ **then**
12:             **break**
13:         **end if**
14:         $\mathcal{P}_{\text{retrieved}} \leftarrow \mathcal{P}_{\text{retrieved}} \cup \{p\}$
15:         chunk_id $\leftarrow \mathcal{M}_{\text{chunk}}[p]$
16:         unique_chunk_ids $\leftarrow$ unique_chunk_ids $\cup \{$chunk_id$\}$
17:     **end for**
18:     $\mathcal{C}_{\text{retrieved}} \leftarrow$ GetChunksFromIDs(unique_chunk_ids)
19:     **return** $\mathcal{P}_{\text{retrieved}}, \mathcal{C}_{\text{retrieved}}$
20: **end function**

---

strictly follows the official code and procedures released by its authors (Zhang et al., 2024b). IRCoT operates on an iterative cycle where the model first generates a reasoning step (a "thought") and then acts upon it. The core of its multi-hop reasoning is guided by the following Chain-of-Thought (CoT) prompt, which instructs the model to generate one reasoning step at a time:

```
You serve as an intelligent assistant, adept at facilitating users through
complex, multi-hop reasoning across multiple documents.  This task is
illustrated through demonstrations, each consisting of a document set
paired with a relevant question and its multi-hop reasoning thoughts.
Your task is to generate one thought for current step, DON'T generate the
whole thoughts at once!  If you reach what you believe to be the final
step, start with "So the answer is:".
```

For **HippoRAG2**, we utilized the official program released by the authors and followed their recommended default hyperparameter settings. This approach builds a knowledge graph from the text, uses Personalized PageRank (PPR) for entity-aware retrieval, and then filters the retrieved facts before the final answer generation. The key hyperparameters, which govern the graph construction and retrieval process, are detailed in Table 2.

Table 2: Hyperparameters for the HippoRAG2 baseline.

| Hyperparameter | Value | Description |
|---|---|---|
| Synonym Threshold | 0.8 | The cosine similarity threshold for merging entity synonyms. |
| Damping Factor (PPR) | 0.5 | The damping factor for the Personalized PageRank algorithm. |
| Temperature | 0.0 | The generation temperature (0.0 ensures deterministic output). |

After retrieving candidate facts, HippoRAG2 employs a filtering step to select the most relevant information. This is guided by the following prompt:

```
You are a critical component of a high-stakes question-answering system...
Your task is to filter facts based on their relevance to a given query...
You must select up to 4 relevant facts from the provided candidate list
```

```
that have a strong connection to the query... The output should be in JSON
format, e.g., {"fact": [["s1", "p1", "o1"], ...]}, and if no facts are
relevant, return an empty list, {"fact": []}...
```

## B.2   A NOTE ON HYPERPARAMETERS

One of the practical advantages of $T^2RAG$ is its simplicity and robustness, as it is largely free of the complex hyperparameter tuning required by other methods. For instance, HippoRAG2 requires careful setting of graph-related parameters (e.g., Synonym Threshold, Damping Factor), and other graph-based methods like Raptor require tuning of clustering parameters. The design of $T^2RAG$ avoids such model-specific tuning, making it easier to deploy and more generalizable across different domains without extensive optimization.

Table 3: Dataset Statistics

| Dataset | PopQA | 2Wiki | Musique | HotpotQA | Story | Medical |
|---|---|---|---|---|---|---|
| # questions | 1000 | 1000 | 1000 | 1000 | 794 | 564 |
| # chunks | 33,595 | 6119 | 11,656 | 9811 | 1266 | 268 |
| # tokens | 2,768,270 | 454,715 | 964,203 | 914,956 | 915,484 | 189,271 |
| # extracted triplets | 398,924 | 65,028 | 127,640 | 124,722 | 22,812 | 5256 |

## B.3   MORE PERFORMANCE RESULTS

### B.3.1   PERFORMANCE ACROSS DIFFERENT LLMS

To provide a robust evaluation of our proposed method, **$T^2RAG$**, we extend the analysis to all six datasets presented in the main content. For each model and method combination, we calculate the averaged **Exact Match (EM)** and **F1 scores**, which are further averaged across all six datasets to produce a single, unified performance score. These methods were evaluated across four powerful language models known for their reasoning capabilities: **GPT-4o-mini**, **Gemini-2.5-flash**, **Qwen-3-Next-Instruct**, and **Gemini-2.5-pro**.

The radar chart in Figure 8 visualizes the aggregated performance across all six datasets, offering compelling evidence to support the central argument presented in our paper.

**$T^2RAG$**, consistently forms the outermost perimeter of the radar chart. This visually demonstrates that our method achieves the highest average EM and F1 score across all four tested language models, from the more compact GPT-4o-mini to the highly capable Gemini-2.5-pro. The performance gap between $T^2RAG$ and other methods is substantial and uniform, highlighting its robustness and state-of-the-art performance. Our main argument posits that $T^2RAG$'s strength lies in its ability to effectively harness the advanced reasoning capabilities of modern LLMs. The chart strongly validates this claim when we compare $T^2RAG$ with other methods. For instance, **IRCoT** (purple line), another method that encourages reasoning, is the second-best performer, closely trailing $T^2RAG$. This suggests that methods explicitly designed to guide the LLM's reasoning process are particularly effective with these models. The hypothesis is further solidified by observing the performance of **HippoRAG2** (pink line). As argued in the main content, methods that relegate the LLM to simpler tasks like filtering may fail to unlock its full potential. The chart shows that HippoRAG2's performance is significantly lower than $T^2RAG$ and IRCoT, and is often comparable to or only slightly better than the standard RAG baseline. This underperformance across a suite of powerful reasoning models supports our hypothesis that its architecture creates a bottleneck, preventing the effective utilization of the LLM's core reasoning strengths. In summary, the aggregated results visualized in the radar chart provide unequivocal support for our central claim. The consistent, chart-topping performance of **$T^2RAG$** across multiple powerful models and diverse datasets confirms that its step-by-step guidance mechanism is uniquely suited to leveraging the sophisticated reasoning abilities inherent in the next generation of large language models.

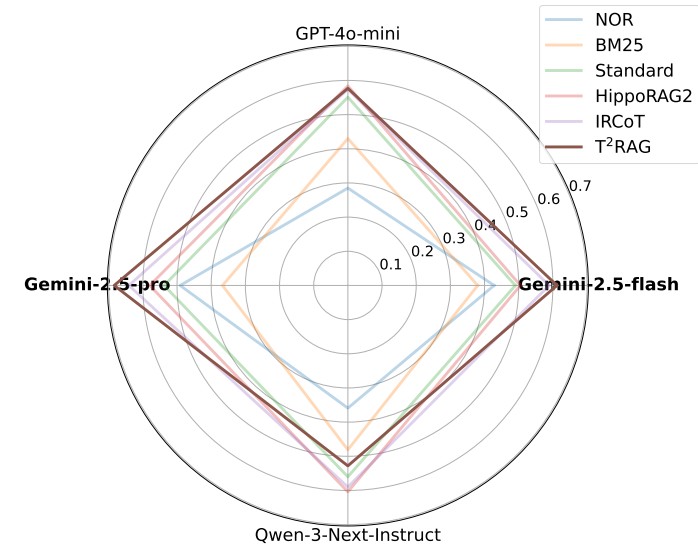

Figure 8: Performance comparison across different LLMs. Reasoning models are in **bold**.

### B.3.2 @REVISED: PERFORMANCE ACROSS HOP COMPLEXITIES@

To analyze performance on longer reasoning chains, we evaluated $T^2RAG$ against strong baselines (HippoRAG2 and IRCoT) on the MuSiQue dataset, stratified by hop count (2, 3, and 4 hops).

Table 4: Performance comparison on the MuSiQue dataset across different hop counts. Best results are bolded.

| Method | 2-hop | | 3-hop | | 4-hop | |
|---|---|---|---|---|---|---|
| | F1 | EM | F1 | EM | F1 | EM |
| HippoRAG2 | 58.79 | 43.82 | 41.90 | 27.22 | **31.14** | **22.29** |
| IRCoT | 43.12 | 33.20 | 31.81 | 22.78 | 13.22 | 7.83 |
| $T^2RAG$ (Ours) | **60.81** | **48.46** | **42.50** | **31.33** | 24.46 | 14.46 |

**Analysis.** As shown in Table 4, $T^2RAG$ consistently achieves State-of-the-Art performance on 2-hop and 3-hop queries. While graph traversal methods like HippoRAG2 exhibit an advantage at extreme hop counts (4-hop) due to global graph connectivity, $T^2RAG$ maintains competitive performance over standard iterative approaches like IRCoT across all complexity levels. Consequently, $T^2RAG$ is particularly recommended for questions requiring reasoning depths of up to 3 hops.

### B.3.3 INDEXING STAGE ANALYSIS

The indexing stage is a one-time, offline process, but its cost can be substantial and even prohibitive for very large corpora. As seen in Figure 9 and Figure 10, datasets like PopQA, 2Wiki, and MuSiQue demand a considerable amount of time and token resources for indexing across all methods. The consumption patterns reveal that **indexing costs are not simply proportional to the raw size of the document corpus.** For instance, the token consumption for RAPTOR's summarization and the triplet extraction for $T^2RAG$ and HippoRAG2 do not scale linearly with the number of documents. This variability likely stems from the **informativeness and density of the source documents.** A document rich with distinct facts will lead to more triplets or more detailed summaries, increasing the computational load, whereas a sparse document will be processed more quickly. This makes the exact indexing cost unpredictable without analyzing the content itself.

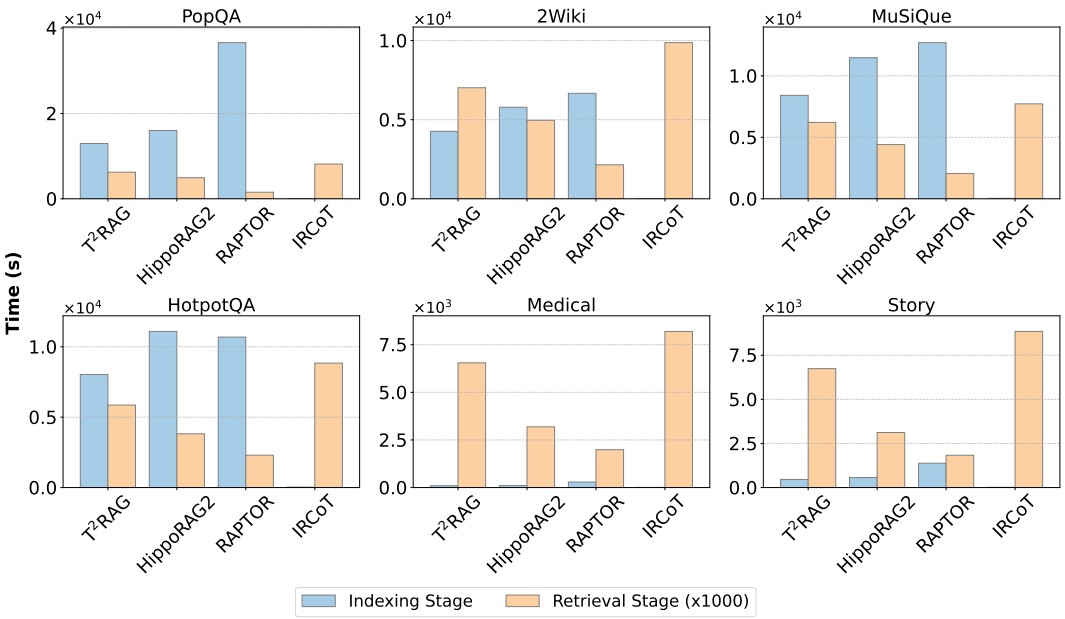

Figure 9: Time consumption at indexing and retrieval stages across all datasets.

### B.3.4 RETRIEVAL STAGE ANALYSIS

The retrieval stage is an online process that occurs for every query, making its efficiency critical for user-facing applications. Our analysis shows that **$T^2$RAG is as efficient as HippoRAG2 during the retrieval stage.** Both methods exhibit similar time and token consumption profiles across all datasets. This is expected, as their retrieval mechanisms are conceptually similar, operating over the graph structures built during indexing.

More importantly, **$T^2$RAG demonstrates a substantial efficiency gain over multi-round RAG methods like IRCoT.** As seen in Figure 10, $T^2$RAG consistently consumes fewer tokens during retrieval than IRCoT across all tested datasets. In some cases, such as the Medical and Story datasets, the reduction in token consumption is over 45%. This efficiency stems from $T^2$RAG's ability to synthesize a direct answer from the retrieved triplets in a single round, avoiding the compounding token costs associated with the iterative query refinement process in multi-round architectures.

Remarkably, **$T^2$RAG often achieves lower, or at least comparable, token consumption than even single-round methods like RAPTOR.** This is particularly evident in datasets like PopQA, Medical, and Story. We attribute this advantage to the nature of the final answer generation. $T^2$RAG generates a concise answer directly from the structured triplets, which minimizes the number of output tokens. Since output tokens are heavily weighted in our consumption metric (multiplied by 4), this concise, triplet-formulated output provides a significant efficiency advantage, leading to an overall reduction in computational cost.

### B.4 MORE EFFICIENCY RESULTS

This section provides a detailed analysis of the time and token consumption of various Retrieval-Augmented Generation (RAG) methods, as illustrated in Figure 9 and Figure 10. The primary goal is to evaluate the computational efficiency of our proposed method, $T^2$RAG, against other established baselines across different stages of the RAG pipeline. The y-axis represents the wall-clock time in seconds required for the indexing and retrieval stages. The retrieval stage time has been scaled by a factor of 1000 to ensure visibility on the chart alongside the much larger indexing times. The y-axis represents the total number of LLM tokens consumed. This is a weighted sum calculated using the formula: **Token Consumption = (#input tokens) + 4 × (#output tokens)**. This weighting reflects the common pricing models of LLM APIs, where generation (output) is typically priced significantly higher (by a factor of 4) than processing (input). As with the time consumption chart, the retrieval

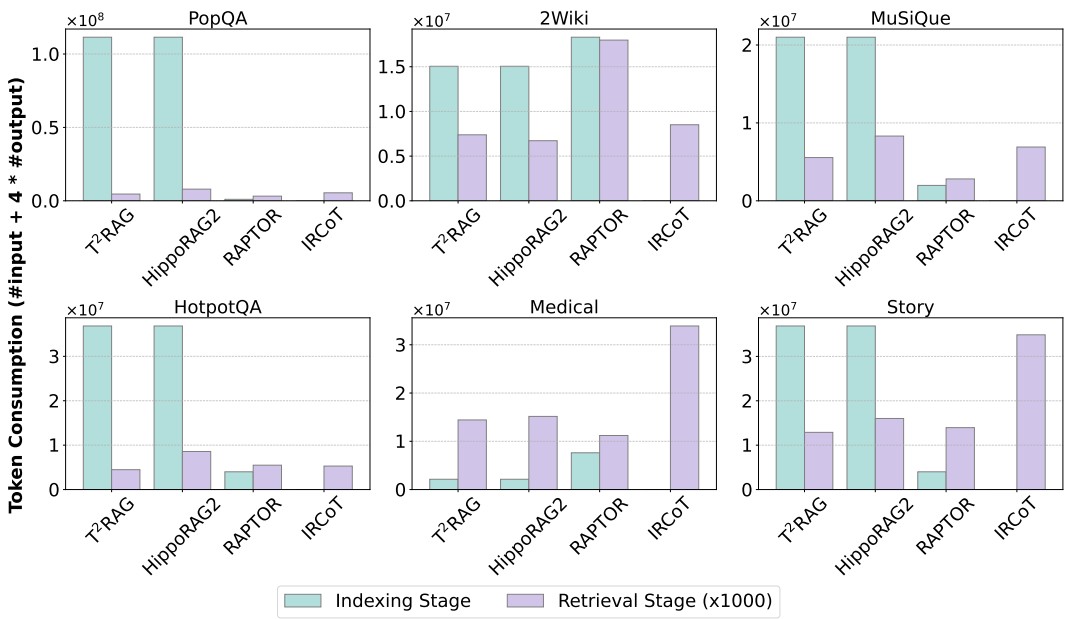

Figure 10: Token consumption at indexing and retrieval stages across all datasets.

stage consumption is scaled by 1000. The x-axis in both figures shows the performance of four methods (T$^2$RAG, HippoRAG2, RAPTOR, and IRCoT) across six distinct datasets.

## B.5 @REVISED: MORE ABLATION STUDY@

To validate the necessity of triplet embeddings, we conducted an ablation study comparing our proposed method against a variant we term **"Chunk-based Slot-Filling."** This baseline utilizes the same iterative slot-filling logic as T$^2$RAG but performs retrieval based on raw chunk embeddings rather than triplet embeddings.

Table 5: Ablation study comparing the proposed T$^2$RAG (Triplet-based) against a Chunk-based baseline. The "Calls" metric indicates the total number of retrieval/LLM interactions. Best results are bolded.

| Method | 2Wiki | | | HotpotQA | | | MuSiQue | | |
|---|---|---|---|---|---|---|---|---|---|
| | **EM** | **F1** | **Calls** | **EM** | **F1** | **Calls** | **EM** | **F1** | **Calls** |
| Chunk-based Baseline | 0.188 | 0.230 | 2000 | 0.301 | 0.397 | 1994 | 0.104 | 0.182 | 1993 |
| **T$^2$RAG (Ours)** | **0.587** | **0.661** | **3355** | **0.565** | **0.700** | **3257** | **0.352** | **0.472** | **3522** |

**Results.** As illustrated in Table 5, removing triplet-based retrieval results in a drastic performance decline, with F1 scores dropping by approximately 40–60% across all datasets.

**Analysis.** We attribute this performance gap to the granularity mismatch inherent in chunk-based retrieval. Without the precise semantic alignment ("hook") provided by triplets, the embedding of a specific decomposed query slot often fails to match the coarser embedding of a large text chunk.

Furthermore, the significantly lower number of "Calls" in the baseline indicates **premature termination**. In the absence of triplets, the system frequently fails to retrieve relevant information to fill the current reasoning slot. Consequently, the iterative loop breaks early, preventing the model from traversing the full reasoning chain required for complex multi-hop questions.

## B.6 More Iteration Results

This analysis examines the average number of retrieval iterations required by T²RAG and IRCoT to answer a query on the 2Wiki dataset, varying the number of retrieved chunks (top-$k$) per iteration.

Table 6: Average Number of Retrieval Iterations vs. top-$k$ on the 2Wiki Dataset.

| topk | T²RAG | IRCoT |
|------|-------|-------|
| 2 | 1.54 | 1.85 |
| 3 | 1.56 | 1.83 |
| 4 | 1.73 | 1.70 |
| 5 | 1.70 | 1.46 |
| 6 | 1.56 | 1.40 |

A key observation from the data is that **T²RAG consistently saves on the number of retrieval iterations compared to IRCoT**, particularly when retrieving fewer documents per step ($k = 2$ or 3). For instance, with $k = 2$, T²RAG requires an average of only 1.54 iterations, whereas IRCoT needs 1.85 iterations—a reduction of approximately 17%. This suggests that T²RAG's method of decomposing a query into structured triplets allows for a more direct and efficient path to resolving the query, requiring fewer rounds of retrieval to gather the necessary context.

The results challenge the simple assumption that retrieving fewer chunks per iteration (a smaller $k$) would necessarily lead to a higher number of total iterations. For T²RAG, the number of iterations remains relatively stable and low, fluctuating between 1.54 and 1.73 without a clear trend. For IRCoT, the relationship is even more complex; as $k$ increases from 4 to 6, the number of iterations surprisingly *decreases* significantly. This indicates that the effectiveness of the retrieved chunks is more important than the sheer quantity. T²RAG's focused retrieval, guided by placeholders in triplets, appears to acquire high-quality context more reliably, making it less dependent on the $k$ value and more efficient overall.

## B.7 @revised: Comparison with More Multi-round Methods@

To evaluate our method against iterative reasoning frameworks, we implemented two agentic baselines: **ReAct** (Yao et al., 2023) and **Self-Ask** (Press et al., 2023).

- **Settings:** We set $k = 5$ for document retrieval and limit the process to a maximum of 3 iterations.
- **Metrics:** We report Exact Match (EM), F1 score, and Total Token usage (calculated as input tokens + 4× output tokens).

Table 7: Performance and efficiency comparison against multi-round baselines (ReAct and Self-Ask) using Gemini-2.5 Flash and GPT-4o-mini. T²RAG achieves superior or competitive accuracy while significantly reducing token consumption.

| Model | Method | 2Wiki | | | HotpotQA | | | MuSiQue | | |
|-------|--------|------|------|--------|------|------|--------|------|------|--------|
| | | EM | F1 | Tokens | EM | F1 | Tokens | EM | F1 | Tokens |
| Gemini 2.5 Flash | ReAct | 0.293 | 0.337 | 395,075 | 0.432 | 0.528 | 611,825 | 0.186 | 0.256 | 431,322 |
| | Self-Ask | 0.083 | 0.148 | 469,714 | 0.310 | 0.391 | 532,756 | 0.147 | 0.229 | 2,195 |
| | **T²RAG** | **0.693** | **0.775** | **366,199** | **0.623** | **0.732** | **606,914** | **0.391** | **0.491** | 336,555 |
| GPT 4o-mini | ReAct | 0.534 | 0.598 | 448,156 | 0.586 | 0.729 | 697,667 | 0.325 | 0.458 | 369,656 |
| | Self-Ask | 0.274 | 0.332 | 504,981 | 0.454 | 0.596 | 324,774 | 0.293 | 0.397 | 459,596 |
| | **T²RAG** | **0.667** | **0.744** | 468,089 | 0.542 | 0.673 | 469,842 | **0.343** | 0.456 | **338,129** |

**Analysis.** As demonstrated in Table 7, T²RAG significantly outperforms both ReAct and Self-Ask across datasets. This advantage is particularly notable in terms of efficiency (token usage). While ReAct demonstrates competitive performance on MuSiQue when using GPT-4o-mini, T²RAG

achieves comparable results with fewer tokens, highlighting its efficiency in handling complex multi-hop scenarios without the high computational overhead of sequential multi-round agents.

## B.8 TRIPLET QUALITY ANALYSIS

### B.8.1 OVERALL PERFORMANCE

We conducted a detailed manual analysis on a set of 50 randomly sampled documents from 2Wiki dataset to quantify the performance of the OpenIE triplet extraction process. The results are summarized in Table 8.

Table 8: Overall performance metrics for the triplet extraction stage.

| Confirmed Docs | Precision | Recall | F1 Score | Avg Triplets/Doc | Avg Entities/Doc |
|---|---|---|---|---|---|
| 50 | 95.9% | 98.2% | 95.0% | 11.6 | 10.5 |

The high Precision (95.9%) and Recall (98.2%) scores demonstrate the overall reliability of the extraction process. This detailed error analysis confirms the robustness of the IE module while also highlighting key areas for future improvement, particularly in reducing the incidence of **missing** and **overmerged** triplets.

### B.8.2 ERROR STATISTICS AND EXAMPLES

Out of the 50 documents, 16 (32.0%) were confirmed to have **perfect extractions** with no errors. The remaining 33 documents (66.0%) contained at least one error. A detailed breakdown of the error types is presented in Table 9.

Table 9: Breakdown of error types across 50 manually annotated documents.

| Error Type | Documents Affected | Percentage |
|---|---|---|
| Missing | 26 | 52.0% |
| Over-merge | 20 | 40.0% |
| Inaccurate | 5 | 10.0% |
| Over-split | 5 | 10.0% |
| Placeholder | 2 | 4.0% |
| Hallucination | 1 | 2.0% |

To provide a clearer understanding of these error categories, we include the following examples:

## B.9 ERROR ANALYSIS

We conducted a comprehensive error analysis on 775 incorrect answer among 1000 samples of MuSiQue dataset to categorize the primary points of weakness in our method. The analysis is done by Gemini-2.5-pro. The results, summarized in Table 12, reveal that the most significant challenge lies in the resolution phase. **"Retrieved correct but wrong resolution"** is the most prevalent error type, accounting for a substantial **46.0%** of all cases. This indicates that even when the system successfully finds the relevant knowledge, it frequently struggles to synthesize it correctly to form the final answer.

The second major bottleneck is **"Missing retrieval,"** which constitutes **31.2%** of the errors, highlighting instances where the necessary facts were never found in the first place. Combined, these two categories represent over 77% of all failures, underscoring that the retrieval and subsequent resolution stages are the most critical areas for improvement. Less frequent, but still notable, issues include **"Hallucination" (12.7%)** and generating an incorrect final answer despite having all correct intermediate steps **(10.0%)**. This analysis strongly suggests that future work should prioritize enhancing the model's ability to not only retrieve the correct atomic pieces of information but also to reason over them accurately.

In addition to solely analyzing one experiment, this toolkit can also facilitate more analysis such as find out why a certain LLM performs much poorer than another.

Table 10: Examples and Descriptions of Triplet Extraction Error Types

| Error Type | Description | Example |
|---|---|---|
| Inaccurate | An element of the triplet is extracted with minor errors. | **Source:** "...Pierre De Geyter, is known for, writing the music of 'The Internationale'."
**Incorrect Extraction:** (Pierre De Geyter, is known for, the music of The Internationale) |
| Over-split | A single cohesive fact is incorrectly split into multiple triplets. | **Incorrect Extractions:** (Denis Sanders, won for, A Time Out of War) + (Denis Sanders, won, Best Short Subject)
**Correct Triplet:** (Denis Sanders, won Best Short Subject for, A Time Out of War) |
| Hallucination | A triplet is generated that is not supported by the source text. | **Source:** "...He is the husband of actress Cate Blanchett."
**Incorrect Extraction:** (Cate Blanchett, is an actress from, Australian) |
| Missing | A key fact present in the text is not extracted. | **Source:** "William I, Elector of Hesse... was the eldest surviving son of Frederick II..."
**Missed Triplet:** (William I, son of, Frederick II) |
| Over-merge | Two or more distinct facts are incorrectly combined into a single triplet. | **Intended Facts:** (James Tuchet, succeeded, James Tuchet) and (James Tuchet, is, 6th Earl of Castlehaven) were incorrectly merged. |
| Placeholder | A generic and uninformative triplet is extracted. | **Source:** "The Trail of the Lonesome Pine may refer to..."
**Incorrect Extraction:** (The Trail..., may refer to, various interpretations) |

Table 11: Distribution of error types across 775 identified failures. The analysis highlights that retrieval and resolution are the primary challenges.

| Error Type | Count | Percentage |
|---|---|---|
| Retrieved correct but wrong resolution | 354 | 46.0% |
| Missing retrieval | 240 | 31.2% |
| Hallucination | 98 | 12.7% |
| All correct but wrong final answer | 77 | 10.0% |
| Unknown / Unclassified | 6 | 0.8% |
| **Total** | **775** | **100.0%** |

## C  RELATED WORK

We group prior efforts into *single-round*, *multi-round*, *graph-enhanced* RAG and *summarization-based* RAG, each adding more interaction or structured reasoning and paving the way for the fine-grained design of T$^2$RAG.

**Single-round RAG.** Classical sparse retrievers such as TF-IDF and BM25 paired with extractive readers perform strongly for open-domain QA (Yang et al., 2019; Nie et al., 2019; Wang et al., 2023a). Dense retrievers such as DPR (Karpukhin et al., 2020) later replaced sparse vectors with learned embeddings, retrieving a fixed top-$k$ set in one pass. *However, answering multi-hop questions often demands the intermediate results to further retrieval, motivating the multi-round techniques that follow.*

**Multi-round RAG.** Due to the missing bridges problem we mentioned in Section 1 more and more works follow a multi-round, training-free paradigm, which enables the LLMs infer the intermediate information thus better retrieve the final answer. Some works focus on the query side. Khot et al. (2023) decompose multi-hop questions into single-hop sub-queries that are solved sequentially. Yao et al. (2023) propose ReAct, interleaving chain-of-thought (CoT) (Wei et al., 2022) steps with search actions issued by the LLM. Similariy, Query2Doc (Wang et al., 2023b) expanding queries into concise triplets to cut token usage while preserving recall. Another line of works relies on the generated intermediate results for next iteration. Beam Retrieval (Zhang et al., 2024a) jointly training an encoder and classifiers to keep multiple passage hypotheses across hops. FLARE (Jiang et al., 2023) forecasts upcoming sentences to decide when fresh retrieval is needed during long-form generation. IRCoT (Trivedi et al., 2023) and ITER-RETGEN (Shao et al., 2023), alternately expanding a CoT and fetching new evidence to answer multi-step questions. Adaptive QA (Xie et al., 2023) create an adaptive framework that picks the simplest effective retrieval strategy according to query complexity. *Despite these advances, few efforts explicitly aim to reduce token costs or number of llm calls during multi-round RAG. Previous methods expand query or generates CoT with long sentences in each round. In contrast, our work minimizes token consumption by formulating query expansions as triplets and simplifying reasoning steps as triplets resolving.*

**Graph RAG.** One major line of research addresses complex QA by structuring knowledge into graphs. Originating in Knowledge Graph QA (KGQA), early methods focused on decomposing queries or performing multi-round, LLM-evaluated traversals from seed nodes (Luo et al., 2024; Sun et al., 2024; Cheng et al., 2024; Mavromatis & Karypis, 2022). The application of this paradigm to general ODQA was popularized by systems that construct a knowledge graph entirely with LLMs and use community detection for retrieval (Edge et al., 2024). Subsequent work has aimed to make this process more efficient. For instance, LightRAG (Guo et al., 2024) introduces a dual-level retrieval system combining graph structures with vector search to improve knowledge discovery. Targeting resource-constrained scenarios, MiniRAG (Fan et al., 2025) builds a heterogeneous graph of text chunks and named entities, enabling lightweight retrieval suitable for Small Language Models. To tackle the common challenge of entity merging, HippoRAG (Gutiérrez et al., 2025a) and HippoRAG2 (Gutiérrez et al., 2025b) create synonym links between similar entity nodes and employs a PageRank (Haveliwala, 1999) algorithm for final node selection. *Despite these advances, a central challenge for Graph RAG remains the costly and error-prone nature of graph construction from unstructured text.*

**Summarization-based RAG.** A distinct but related approach focuses on building hierarchical summarization trees rather than explicit graphs. These methods aim to capture information at varying levels of abstraction. For example, Raptor (Sarthi et al., 2024) constructs a summary tree by recursively clustering document chunks and summarizing the content within each cluster to create new, more abstract retrieval units (Wu et al., 2023). Aiming to capture more detailed contextual information, SireRAG (Zhang et al., 2024b) creates a "relatedness tree" by summarizing fine-grained propositions that share the same entities. *However, these summarization-based methods often incur high computational costs during the indexing phase and risk losing the fine-grained, factual details that are essential for precise factoid QA.*

# D LIMITATIONS

## D.1 OVERVIEW

Although our method achieves state-of-the-art performance with a simple design, it is not without limitations. **Experimentally**, we limited our multi-round methods to 3 iterations to match the complexity of the datasets and ensure a fair efficiency comparison; we also did not have the resources to test on other embedding models especially LLM-based ones, re-rankers or large external knowledge graphs (e.g., Wikipedia KG (Hertling & Paulheim, 2018)). Our evaluation is also limited to the black-box and end-to-end one which may lack explanability without the recall score of chunks. **Methodologically**, our approach is highly dependent on the quality of the triplet extraction. While higher-quality sources can be used, simple triplets may not adequately represent complex knowledge like many-to-many relationships, a challenge that could be addressed with hypergraph modeling (Luo et al., 2025) in future work. Besides, the efficiency of triplet extraction can be further improved beyond the classic OpenIE pipeline. Developing these methods needs efforts from information extraction (Grishman, 2015) area. **Finally**, regarding scalability, building the index from a very large corpus is token-intensive. However, our method is very efficient when using a pre-existing triplet database. This design also makes it inherently suitable for evolving knowledge bases, as new triplets are independent to previous ones thus they can be added incrementally, offering a significant advantage over static Graph RAG approaches (Zhang et al., 2025).

## D.2 @REVISED: EXPERIMENTAL FAIRNESS@

**Retrieval Volume Balance.** A potential concern in comparing multi-round methods against direct chunk-retrieval baselines is the "information imbalance," where iterative methods may access a larger volume of text. To address this, our analysis in Figure 7 utilizes **Effective Top-K** (defined as $k \times$ average number of iterations) for the X-axis when plotting multi-round methods. This normalization ensures we compare methods based on the total volume of information retrieved. The results demonstrate that $T^2RAG$ maintains consistent performance advantages across different Effective Top-K values, confirming that its superiority stems from enhanced reasoning and retrieval precision rather than simply accessing more context. For the main results table, we adhere to the evaluation protocols established in HippoRAG2 (Gutiérrez et al., 2025b), where variation in retrieved content volume is treated as an inherent characteristic of the respective methodologies.

## D.3 @REVISED: EDGE CASE HANDLING@

We explicitly define the system's behavior in the rare instance where no searchable triplets are identified during an iteration. If the query decomposition or refinement step fails to generate valid triplets, the iterative retrieval loop terminates immediately. The system then proceeds directly to the final answer generation phase utilizing the context retrieved up to that point. Empirically, valid natural language queries typically contain sufficient semantic structure to form at least one searchable triplet, making this edge case extremely uncommon. A "retry mechanism" where the LLM could re-attempt decomposition or resolution upon failure could be helpful. While this could improve robustness, it introduces a trade-off between performance and efficiency.

**Question**: Which film has the director born earlier, God's Gift To Women or Aldri Annet Enn Brak?

Query Decomposition

**Q:** Think step by step about what information is needed to answer this question. Form triples in the format: subject | predicate | object. Use "?" as placeholder for unknown entities…
**A:**
**Searchable Triplets:** God's Gift To Women, is directed by, ?DirectorA ; Aldri Annet Enn Brak, is directed by, ?DirectorB ;
**Fuzzy Triplets:** ?DirectorA, was born in, ?birthYearA ; ?DirectorB, was born in, ?birthYearB

Iteration 1

**Q:** Use the context passages and propositions to solve any "?" placeholders with as much detail as possible, grounding your answers in the passage content.
**Retrieved Triplets**: Aldri annet enn brak was directed by Edith Carlmar, God's Gift to Women directed by Michael Curtiz, God's Gift to Women was completed as musical film, Aldri annet enn brak is 1954 Norwegian comedy-drama film, God's Gift to Women starring Joan Blondell, Aldri annet enn brak has English title Nothing but trouble, God's Gift to Women is 1931 American pre-Code romantic musical comedy film … **(18 propositions in total)**
**Retrieved Chunks**: Title: Altid ballade … Title: God's Gift to Women … Title: Aldri annet enn brak … Title: Edith Carlmar (Edith Mary Johanne Mathiesen) (15 November 1911 - 17 May 2003) Title: Stanley Kwan
**A:**
**Newly Resolved triplets:** God's Gift To Women, is directed by, Michael Curtiz ; Aldri Annet Enn Brak, is directed by, Edith Carlmar ; Edith Carlmar, was born in, 1911
**Newly searchable chunks:** Michael Curtiz, was born in, ?birthYearA ;

Iteration 2

**Q:** Use the context passages and propositions to solve any "?" placeholders with as much detail as possible, grounding your answers in the passage content.
**Retrieved Triplets**: Michael Curtiz born as Mih1ly Kertsz, Michael Curtiz was Hungarian-born American film director, Michael Curtiz was born on December 24, 1886, Michael Curtiz made stars of Bette Davis, The Vagabond King directed by Michael Curtiz, Bright Leaf directed by Michael Curtiz…**(32 propositions in total)**
**Retrieved chunks**: Title: Michael Curtiz Michael Curtiz oMichael Curtiz (born Man Kammsert)1886-19051Mih1ly Kertz (1905); December 24, 1886 April 11, 1962) was a Hungarian-born American film director… Title: Bright Leaf… Title: Altid ballade…Title: The Vagabond King… Title: JdJds…
**A:**
**Newly Resolved Triplets:** Michael Curtiz, was born in, 1886
**Newly Searchable Triplets**: None. **(all triplets resolved and the iteration is terminated)**

Final Answering

**Q:** Based on the following triplets, please answer the following question.
**Total Resolved Triplets**: God's Gift To Women, is directed by, Michael Curtiz ; Aldri Annet Enn Brak, is directed by, Edith Carlmar ; Michael Curtiz, was born in, 1886 ; Edith Carlmar, was born in, 15 November 1911.
**A:** God's Gift To Women

Figure 11: An example of T$^2$RAG QA. To answer the question, we need intermediate facts about Michael Curtiz (marked by yellow and Edith Carlmar (marked by red), which are not reflected in the question.

# E  CASE STUDY

## E.1  ONE CASE FROM OUR METHOD

We offer a full log of T$^2$RAG during our experiment running in Figure 11.

This case study showcases the effectiveness of resolving the complex comparative query in 2 retrieval iterations. The system successfully decomposed the query into 4 necessary triplets (two directors, two birth years) and retrieved context only by the searchable ones. By identifying both directors (Michael Curtiz, Edith Carlmar) and their birth years (1886, 1911) from the triplet DB or initial set of chunks, it bypassed the need for further retrieval rounds. This immediate and complete information acquisition demonstrates the power of T$^2$RAG's query decomposition and high-quality triplet-based retrieval.

## E.2  @REVISED: COMPARISON WITH BASELINES@

We analyzed specific instances where T$^2$RAG succeeds while strong query planning baselines fail. A representative example is the multi-hop question: *"Who is the child of the performer of the song Me And Bobby McGee?"*

**Mechanism Analysis: Structured Variable Locking.**  T$^2$RAG's success in this scenario is attributed to a mechanism we term **Structured Variable Locking**.

1. **Decomposition:** The system decomposed the query into rigid logical constraints:

$$(\text{Me And Bobby McGee}, \text{performer}, \text{?performer})$$

$$(\text{?performer}, \text{child}, \text{?child})$$

2. **Step 1 Resolution:** The system successfully resolved ?performer to "Roger Miller."

Table 12: Case study comparing error modes between baselines and T$^2$RAG. T$^2$RAG succeeds due to its variable-locking mechanism.

| Method | Answer | Verdict | Error Mode / Mechanism |
|---|---|---|---|
| HippoRAG | "Not specified" | Failure | **Graph Disconnection:** The traversal failed to immediately identify a link between the song node and a child node, leading the planner to incorrectly conclude the data did not exist. |
| IRCoT | "Info not available" | Failure | **Premature Abandonment:** While the Chain-of-Thought correctly identified the performer, the agent abandoned the secondary retrieval step for the child after a single failed retrieval attempt. |
| **T$^2$RAG** | **"Dean Miller"** | **Success** | **Structured Variable Locking:** The persistence of the unbound placeholder `?child` forced the system to continue iterating until the specific relationship was resolved. |

3. **Forced Iteration:** Unlike IRCoT, which halts upon a transient retrieval failure, T$^2$RAG adheres to the logical constraint: *"I have resolved Roger Miller, but the slot for* `?child` *remains empty."* This "empty slot" forces the system into subsequent iterations of targeted retrieval specifically for "Roger Miller's family/children," eventually locating the fact that Dean Miller is his son.

## F    PROMPTS

We provide all prompt templates we used at retrieval stage, namely structured query decomposition, triplet resolving and final answering. These are prompts used in LLM$_{\text{Decompose}}$, LLM$_{\text{Resolve}}$, LLM$_{\text{Answer}}$, respectively. $\{\cdot\}$ represents the content needed to be replaced by the original question, intermediate generated triplets, or retrieved propositions and chunks.

## G    @REVISED: DISCUSSION@

### G.1    PERFORMANCE ON LARGE CORPUS

**Embedding Quality and Noise.**    One potential concern regarding triplet embeddings is the introduction of noise compared to graph structures. However, we posit that triplets actually **reduce** noise compared to chunk-level embeddings.

- **Granularity:** A chunk embedding averages multiple facts into a single vector, often diluting specific details. In contrast, triplet embeddings isolate specific relationships (`Subject-Predicate-Object`), resulting in cleaner and more precise retrieval signals.

- **Modern Encoders:** With the rapid development of LLM-based embedding models, the semantic quality of triplet representations is high, and the dimensionality remains manageable. Quality can be further enhanced by scaling the number of dimensions or model parameters (Muennighoff et al., 2023).

**FAISS Efficiency.**    Regarding approximation errors in retrieval:

- The approximation error in modern Approximate Nearest Neighbor (ANN) libraries, such as FAISS, is theoretically bounded.

**Structured Query Decomposition**

You are tasked with reasoning about a question and extracting the necessary knowledge triples to answer it.
**Instructions**:
1. Think step by step about what information is needed to answer this question
2. Form triples in the format: subject | predicate | object
3. Use "?" as placeholder for unknown entities
4. For comparative questions involving multiple entities, use distinct placeholders like ?entityA, ?directorA, ?directorB
5. Extract multiple triples if the question requires complex reasoning
**Examples**:
- Question: "What is the capital of France?" Reasoning: To answer this, I need to know what France's capital is. Triple: France | has capital | ?
- Question: "Who directed the movie that won Best Picture in 2020?" Reasoning: To answer this, I need to know which movie won Best Picture in 2020, and who directed that movie. Triples: ? | won Best Picture | 2020 ? | is directed by | ?
- Question: "Which film whose director was born first, MovieA or MovieB?" Reasoning: To answer this, I need to know the director of each movie, and the birth year of each director to compare them. Triples: MovieA | is directed by | ?directorA MovieB | is directed by | ?directorB ?directorA | was born in | ? ?directorB | was born in | ?
Now analyze this question:
**Question**: {query}
Provide your response in this format:
**Reasoning**: [Your step-by-step reasoning about what information is needed]
**Triples**: [List each triple on a new line in format: subject | predicate | object]

**Triplets Resolving**

Example: Context Propositions: {context propositions}
Fully Resolved Clue 1: Subject: Lothair II Predicate: has mother Object: Ermengarde of Tours
Newly Searchable Clue 1: Subject: Ermengarde of Tours Predicate: died on Object: ?
—
Now apply the same process to the following clues: Use the context passages and propositions to resolve any '?' placeholders with as much detail as possible, grounding your answers in the passage content. Instructions:
1. For searchable clues (one '?'), replace '?' with the correct entity to fully resolve it, including any relevant attributes.
2. For fuzzy clues (multiple '?'), generate a Newly Searchable Clue by replacing one of the placeholders with the correct entity, including any relevant context.
Original Query: {query}
Searchable Clues: {searchable clues text}
Fuzzy Clues: {fuzzy clues text}
Context Passages: {context passages}
Context Propositions: {context propositions}
Previous Resolved Clues: {resolved clues context}
Return two lists in this format:
Fully Resolved Clue 1: Subject: ... Predicate: ... Object: ...
Fully Resolved Clue 2: Subject: ... Predicate: ... Object: ...
Newly Searchable Clue 1: Subject: ... Predicate: ... Object: ...
Newly Searchable Clue 2: Subject: ... Predicate: ... Object: ...
(Continue numbering accordingly)

> **Final Answering**
>
> Based on the reasoning clues, please answer the following question.
> Question: {query}
> Key Reasoning Clues: {total resolved clues + remaining searchable clues}
> Instructions:
> 1. Analyze the question step by step
> 2. Use the reasoning clues to understand what information is needed
> 3. Provide ONLY a concise answer
> Answer format requirements:
> - For WH questions (who/what/where/when): Provide the exact entity, date, full name, or full place name only
> - For yes/no questions: Answer only "yes" or "no"
> - No explanations, reasoning, or additional text
> - One entity or fact only
> Answer:

- Noise from embedding retrieval is a universal challenge affecting all dense retrieval methods—including those used to construct graphs or retrieve chunks in baselines—and is not specific to T$^2$RAG.

### G.2 RELATION TO GRAPH RAGS

A key distinction of our method is shifting the graph construction phase from **offline indexing** to **online inference**. This approach does not discard structural information but rather adapts how it is utilized.

**Implicit Inference-Time Graph Construction.** While recent works have explored explicit graph construction during inference, few investigate **implicit graph construction**—providing necessary nodes (entities) and edges (relations) at the final answering stage without pre-hardwired connections. By allowing the LLM to construct necessary graph structures internally within its context window, T$^2$RAG offers distinct advantages:

- **Handling Ambiguity:** The model can naturally handle entity ambiguity and diverse reasoning structures (chains, trees, or DAGs) dynamically based on the query, avoiding reliance on rigid, pre-defined schemas.
- **Leveraging Reasoning Models:** Our experiments indicate that T$^2$RAG benefits significantly from the enhanced reasoning capabilities of state-of-the-art models (e.g., Gemini-2.5-flash). In contrast, methods relying on rigid graph pre-processing (such as HippoRAG2 (Gutiérrez et al., 2025b) or RAPTOR (Sarthi et al., 2024)) may not fully capitalize on these gains, as "hard" graph structures can constrain the "soft" reasoning power of the LLM.

**The "Locality Assumption" and Retrieval Equivalence.** We posit that complex offline graph construction is often unnecessary for effective retrieval due to the **locality assumption**, particularly for questions within 3 hops.

1. In Knowledge Graphs, a text chunk typically forms a star graph or a densely connected cluster.
2. Traversing a pre-built subgraph yields a similar information set to our method, which retrieves all triplets associated with a relevant chunk.

T$^2$RAG aggregates these triplets and allows the LLM to filter or combine them using the decomposed query. This achieves a "PageRank-style" filtering effect but transfers the computational burden to the highly capable LLM during inference, rather than the indexing stage.

**Efficiency and Performance Trade-off.** T$^2$RAG maximizes efficiency compared to GraphRAG and performance compared to multi-round RAG. It offers a robust balance, particularly for 1-2

hop QA applications, while retaining the potential for deeper reasoning as base models continue to improve.

