# OpenReview forum: "Beyond Chunks and Graphs: Retrieval-Augmented Generation through Triplet-Driven Thinking"
_ICLR.cc/2026/Conference — ICLR 2026 Conference Withdrawn Submission_

### Official Review · Reviewer_AQoT · 2025-10-28

**Soundness:** 3
**Presentation:** 2
**Contribution:** 3
**Rating:** 6
**Confidence:** 4

**Summary:**

This paper proposes $T^2RAG$, a lightweight and efficient retrieval-augmented generation framework that uses a graph-free knowledge base of atomic triplets. $T^2RAG$ decomposes questions into searchable triplets with placeholders and iteratively resolves them via evidence retrieval, reducing reliance on costly multi-round interactions or complex graph construction. Experiments show it outperforms state-of-the-art RAG methods, offering a more effective and efficient solution for knowledge-intensive reasoning.

**Strengths:**

* Using triplet with placeholder to represent the required information is new and interesting to me
* The experiments are extensive

**Weaknesses:**

* The triplet update itself is not complicated, but in Step 2.3, the use of mathematical notation makes the process significantly harder to follow.
* T^2RAG is actually a multi rould RAG, but only one multi round baseline is compared, maybe some more multi round baselines should be added.

**Questions:**

* In section 4.3, line 262, the authors convert the triplet into query by simply concatenating the elements without the placeholder, but this query will be quite different from the normal ones, will the dense retriever perform suboptimally due to out of domain?
* I do not quite understand why using triplet can reduce the number of llm calls. For each retrieval, the method still requires an LLM call to update the triplet, and other approaches can similarly update the next subquery after each retrieval.
* Instructing the LLM to generate triplet seems to be difficult, how does small models like Llama 8B and Qwen 7B performs

---

> ### Author Response · Authors · 2025-11-24
>
> We sincerely thank Reviewer AQoT for their constructive feedback, particularly regarding the clarity of our notation and the need for broader multi-round baseline comparisons. We have addressed each point below.
>
> > W1: The triplet update itself is not complicated, but in Step 2.3, the use of mathematical notation makes the process significantly harder to follow.
> >
>
> We thank the reviewer for this feedback. We use the mathematical notation to highlight the reasoning process of triplet-driven thinking. Figure 2 illustrates our core method without the use of mathematical notation. In the revised manuscript, we have simplified the main text description of Step 2.3 to focus on the intuitive logic (as illustrated in Figure 2) and move the formal mathematical definitions to the Appendix A.
>
> > W2: T2RAG is actually a multi rould RAG, but only one multi round baseline is compared, maybe some more multi round baselines should be added.
> >
>
> We thank the reviewer for this valuable suggestion. To provide a comprehensive comparison, we have implemented and evaluated two additional widely adopted multi-round methods: **ReAct** [2] and **Self-Ask** [1].
>
> - **Experimental Settings:**
> • $k=5$, Max Iterations $=3$.
> • **ReAct:** Uses `Thought`, `Action` (Search), and `Observation` loop.
> • **Self-Ask:** Decomposes complex queries into simpler follow-up questions.
>
>     We report the EM, F1, Totoal tokens = input tokens + 4 * output tokens as follows.
>
>     **Results:**
>
>     | Gemini-2.5 flash |  |  |  |  |  |  |  |  |  |
>     | --- | --- | --- | --- | --- | --- | --- | --- | --- | --- |
>     | Method | 2Wiki EM | 2Wiki F1 | 2Wiki Tokens | Hotpot EM | Hotpot F1 | Hotpot Tokens | Musique EM | Musique F1 | Musique Tokens |
>     | ReAct | 0.293 | 0.337 | 395,075 | 0.432 | 0.528 | 611,825 | 0.186 | 0.256 | 431,322 |
>     | Self-Ask | 0.083 | 0.148 | 469,714 | 0.31 | 0.391 | **532,756** | 0.147 | 0.2 | **292,195** |
>     | T2RAG | **0.693** | **0.775** | **366,199** | **0.623** | **0.732** | 606,914 | **0.391** | **0.491** | 336,555 |
>
>     | GPT-4o-mini |  |  |  |  |  |  |  |  |  |
>     | --- | --- | --- | --- | --- | --- | --- | --- | --- | --- |
>     | Method | 2Wiki EM | 2Wiki F1 | 2Wiki Tokens | Hotpot EM | Hotpot F1 | Hotpot Tokens | Musique EM | Musique F1 | Musique Tokens |
>     | ReAct | 0.534 | 0.598 | **448,156** | 0.586 | 0.729 | 697,667 | 0.325 | **0.458** | 369,656 |
>     | Self-Ask | 0.274 | 0.332 | 504,981 | 0.454 | 0.596 | **324,774** | 0.293 | 0.397 | 459,596 |
>     | T2RAG | **0.667** | **0.744** | 468,089 | 0.542 | 0.673 | 469,842 | **0.343** | 0.456 | **338,129** |
>
>     **Analysis**: T2RAG significantly outperforms both ReAct and Self-Ask across datasets, particularly in efficiency (tokens). While ReAct shows competitive performance on MuSiQue with GPT-4o-mini, T2RAG achieves comparable results with fewer tokens, highlighting its efficiency in complex multi-hop scenarios.
>
>
> > Q1: In section 4.3, line 262, the authors convert the triplet into query by simply concatenating the elements without the placeholder, but this query will be quite different from the normal ones, will the dense retriever perform suboptimally due to out of domain?
> >
>
> We thank the reviewer for raising this concern about retrieval distribution.
> We clarify that "searchable triplets" (e.g., `Subject + Predicate`) typically act as highly effective **keyword queries**. Dense retrievers (like Contriever or OpenAI embeddings) are trained on massive corpora that include diverse query formats, and they generally exhibit high tolerance for keyword-style inputs.
> Furthermore, since the triplet embeddings are derived from LLMs trained on next-token prediction, they maintain semantic alignment with the text chunks (which are essentially the natural language realization of those triplets). Our ablation studies (Standard T2RAG) show that the structure of the triplet actually enhances retrieval precision compared to raw chunk retrieval, rather than degrading it.

---

> ### Author Response · Authors · 2025-11-24
>
> > Q2: I do not quite understand why using triplet can reduce the number of llm calls. For each retrieval, the method still requires an LLM call to update the triplet, and other approaches can similarly update the next subquery after each retrieval.
> >
>
> We thank the reviewer for this opportunity to clarify the efficiency gains.
> The reduction in LLM calls and computational cost comes from two factors:
>
> - As shown in Table 4 (specifically when $k$ is small), leveraging extracted triplets provides a precise "hook" into the data. This allows T2RAG to resolve specific logical steps faster than chunk-based methods (like IRCoT), which often suffer from "retrieval drift" (distracted by irrelevant retrieved information and generating wrong hooks) and require more iterations (and thus more LLM calls) to locate the correct information.
> - The primary efficiency gain is in **output tokens**. In each iteration, T2RAG outputs structured, concise updates (filling a slot) rather than generating verbose Chain-of-Thought paragraphs. Since API costs and latency are heavily driven by output generation, this results in significant efficiency improvements.
>
> > Q3: Instructing the LLM to generate triplet seems to be difficult, how does small models like Llama 8B and Qwen 7B performs
> >
>
> We thank the reviewer for this insightful question. We agree that T2RAG relies on the reasoning and instruction-following capabilities of the LLM. Our design philosophy aims to leverage the emerging reasoning power of modern models (e.g., GPT-4o, Gemini 1.5) to perform structured reasoning with minimum tokens. Smaller models (e.g., 7B/8B parameters) indeed struggle more with complex constraint satisfaction and may hallucinate structure. However, we agree with that fine-tuning these smaller models on triplet-reasoning tasks is a promising direction to bridge this gap. In the revised manuscript, we will include a discussion and trend analysis showing performance across model sizes to make the bounds of our method transparent.
>
> We once again thank the reviewer for their helpful comments and look forward to improving the paper based on this feedback.
>
> ## References
>
> 1. Measuring and Narrowing the Compositionality Gap in Language Models. In EMNLP 2023
> 2. React: Synergizing reasoning and acting in language models. In ICLR 2022

---

### Official Review · Reviewer_pP8e · 2025-10-31

**Soundness:** 2
**Presentation:** 3
**Contribution:** 2
**Rating:** 4
**Confidence:** 3

**Summary:**

The paper proposes T2RAG, a retrieval-augmented generation framework that replaces chunk-level retrieval with a graph-free store of atomic triplets. An offline stage distills a corpus into triplets and embeds them with FAISS for fast lookup. At inference time, an LLM decomposes the user question into placeholder triplets, iteratively resolves missing arguments by retrieving candidate triplets, and finally answers the question using the resolved evidence. Across six datasets, the authors report performance gains over selected multi-round and graph-based RAG baselines while reducing retrieval costs.

**Strengths:**

1.The work proposes triplet-driven thinking to reduce token footprint and simplify graph dependencies in the RAG pipeline.

2.The split between offline distillation (triplet extraction/embedding) and online iterative resolution is clean and easy to reproduce, with a plausible path to system engineering and deployment. The placeholder-resolution idea encourages explicit evidence grounding rather than opaque, multi-turn prompts; this has the potential to reduce uncontrolled drift.

3.The paper claims consistent improvements on multiple datasets along with cost reductions, suggesting that lighter-weight semantic atoms can be competitive with heavier chunk or graph pipelines.

**Weaknesses:**

1.The settings of experiments appear to resolve to entity answers or facts naturally expressible as single triplets. This inherently benefits triplet retrieval and makes direct comparisons to multi-round RAG and GraphRAG less fair. The paper should include tasks where answers are compositional, multi-hop, or non-entity (rationales, procedural steps) to validate generality.

2.The placeholder triplet decomposition and iterative resolution are described verbosely, but the novelty claim would be stronger with: (a) ablations comparing against simpler query rewriting + re-ranking loops; (b) a version that uses slot-filling over chunks without pre-extracted triplets; and (c) error-driven analyses showing why placeholders materially outperform strong query planners.

3.Ablation results indicate that raw chunks still matter during retrieval, and that intermediate LLM calls over these chunks increase token usage and hallucination risk. This undercuts the central claim that “replacing chunks with triplets” consistently reduces cost and risk.

4.There is no robustness study for incorrect or incomplete query-triplet decomposition. Since downstream retrieval depends on these slots, small decomposition errors could derail performance on complex questions.

5.Only a few representatives of multi-round and GraphRAG are included. More representative baselines should be added to support the “state-of-the-art” claim. Also report exact prompting/cost budgets for each baseline to ensure fairness.

6.Triplet-first retrieval may struggle on compositional, multi-hop, or explanatory tasks where answers are not atomic entities, which limits the practical value of T2RAG.

7.Minor typos: Line 280: “and and.” Line 415: “Figure 5” should be “Table 2”.

**Questions:**

Please refer to Weaknesses.

---

> ### Author Response · Authors · 2025-11-24
>
> We thank the reviewer for their insightful feedback and the opportunity to clarify these important points.
>
> > W1: The settings of experiments appear to resolve to entity answers or facts naturally expressible as single triplets. This inherently benefits triplet retrieval and makes direct comparisons to multi-round RAG and GraphRAG less fair. The paper should include tasks where answers are compositional, multi-hop, or non-entity (rationales, procedural steps) to validate generality.
> >
>
> We agree with the first point but this setting is non-trivial. The answers or facts can not be achieved from single triplets but needs multiple reasoning steps. We have clarified that our work specifically focuses on **Factoid QA** (Line 115), a domain where even current SOTA methods (multi-round RAG and GraphRAG) have not yet saturated performance (often <70% F1 on complex benchmarks).
>
> - **Multi-hop Complexity:** We explicitly test on **MuSiQue** (2-4 hops), **HotpotQA**, and **2WikiMultihopQA**. These are standard, rigorous benchmarks used by major baselines (HippoRAG, IRCoT, LightRAG [1,2,3,4]).
> - **Scope:** While we agree compositional/procedural tasks are important, they often require distinct evaluation metrics (e.g., human eval or model-based scoring) compared to the exact-match precision required for Factoid QA. Addressing the "elemental" challenge of retrieving precise multi-hop facts is a prerequisite for reliable complex reasoning. Our method aims to maximize retrieval precision for these foundational factoid queries.
>
> > W2: novelty claim would be stronger with: (a) ablations comparing against simpler query rewriting + re-ranking loops; (b) a version that uses slot-filling over chunks without pre-extracted triplets; and (c) error-driven analyses showing why placeholders materially outperform strong query planners.
> >
>
> We have performed the suggested analyses and ablations.
>
> **(a) Query Rewriting Baseline:** We utilized **IRCoT** as our query-rewriting baseline. As detailed in our setup, IRCoT uses an iterative "Thought-Action" loop to rewrite queries and retrieve new context. T2RAG significantly outperforms this standard rewriting approach (see Main Table).
>
> **(b) Ablation: Slot-Filling without Triplets (Standard T2RAG):**
> We implemented "Standard T2RAG," which uses the same slot-filling logic but retrieves based on raw chunk embeddings rather than triplet embeddings.
>
> | **Method** | **2Wiki EM** | **2Wiki F1** | **2Wiki Calls** | **HotpotQA EM** | **HotpotQA F1** | **HotpotQA Calls** | **Musique EM** | **Musique F1** | **Musique Calls** |
> | --- | --- | --- | --- | --- | --- | --- | --- | --- | --- |
> | **T2RAG** | 0.587 | 0.661 | 3355 | 0.565 | 0.700 | 3257 | 0.352 | 0.472 | 3522 |
> | **Standard T2RAG** | 0.188 | 0.230 | 2000 | 0.301 | 0.397 | 1994 | 0.104 | 0.182 | 1993 |
>
> **Result:** Performance drops drastically (~40-60% F1 reduction).
> **Analysis:** Without the precise "hook" of a triplet, the embedding granularity mismatch between the decomposed query slot and a large text chunk causes retrieval failures. The lower call count in Standard T2RAG indicates premature termination: the system fails to find relevant information to fill the current slot, causing the iterative chain to break early.
>
> **(c) Error-Driven Analysis:**
> We analyzed cases where T2RAG succeeds while strong query planners fail.
>
> **Question:** *"Who is the child of the performer of song Me And Bobby McGee?"*
>
> | **Method** | **Answer** | **Verdict** | **Error Mode** |
> | --- | --- | --- | --- |
> | **HippoRAG** | "Not specified in the text" | **Failure** | The graph traversal failed to immediately link the song to a child node, leading the planner to incorrectly conclude the data did not exist. |
> | **IRCoT** | "Information not available" | **Failure** | The chain-of-thought identified the performer but **give up** second retrieval step for the child after one failure retrieval. |
> | **TRAG** | **"Dean Miller"** | **Success** | The placeholder `?child` remained "unbound," forcing the system to continue iterating until the specific relationship was found. |
>
> **T2RAG succeeded due to Structured Variable Locking:**
>
> 1. **Decomposition:** TRAG broke the query into rigid logic: `(Me And Bobby McGee, performer, ?performer)` and `(?performer, child, ?child)`.
> 2. **Step 1 Resolution:** It successfully resolved `?performer` to **Roger Miller**.
> 3. **Forced Iteration:** Unlike the IRCoT, TRAG **cannot** discard the second triple. The system maintained the logical constraint: *I have Roger Miller, but I still have an empty slot for `?child`.* This "empty slot" forced the system into next **different iterations** of targeted retrieval specifically for "Roger Miller's family/children," eventually locating the fact that **Dean Miller** is his son.

---

> ### Author Response · Authors · 2025-11-24
>
> > W3: Ablation results indicate that raw chunks still matter during retrieval, and that intermediate LLM calls over these chunks increase token usage and hallucination risk. This undercuts the central claim that “replacing chunks with triplets” consistently reduces cost and risk.
> >
>
> We apologize for any confusion. We do **not** claim to replace chunks with triplets for the *reading* stage, but rather for the *indexing and retrieval* stage.
>
> - T2RAG uses triplets as precise **retrieval indices (hooks)**. Once a triplet is retrieved, we map it back to its parent text chunk to enhance relevant details for the LLM to read.
> - The efficiency gain comes from the **resolution** phase. By validating logic against triplets first, we reduce the volume of irrelevant chunks fed to the LLM context window compared to standard top-k chunk retrieval, thereby reducing output generation tokens (as shown in Table X) and hallucination risk.
>
> > W4: There is no robustness study for incorrect or incomplete query-triplet decomposition. Since downstream retrieval depends on these slots, small decomposition errors could derail performance on complex questions.
> >
>
> We acknowledge that decomposition errors can cascade. Our error analysis (Table 8) indicates that "Retrieved correct but wrong resolution" accounts for 46% of errors, while "Missing retrieval" is 31.2%. However, T2RAG’s iterative nature offers inherent robustness compared to one-shot decomposition. If a decomposition is slightly off, the "fuzzy matching" in the retrieval stage often surfaces relevant triplets that allow the LLM to self-correct the reasoning path in subsequent iterations.
>
> > W5: Only a few representatives of multi-round and GraphRAG are included. More representative baselines should be added to support the “state-of-the-art” claim. Also report exact prompting/cost budgets for each baseline to ensure fairness.
> >
> - HippoRAG2 is currently the SOTA in these datasets we used, considering both performance and efficiency in a popular benchmark paper[5].
> - We added two more multi-round method adapted from SelfAsk [6] and React [7] as follows.
>     - **Experimental Settings:**
>     • $k=5$, Max Iterations $=3$.
>     • **ReAct:** Uses `Thought`, `Action` (Search), and `Observation` loop.
>     • **Self-Ask:** Decomposes complex queries into simpler follow-up questions.
>
>         We report the EM, F1, Totoal tokens = input tokens + 4 * output tokens as follows.
>
>         **Results:**
>
>         | Gemini-2.5 flash |  |  |  |  |  |  |  |  |  |
>         | --- | --- | --- | --- | --- | --- | --- | --- | --- | --- |
>         | Method | 2Wiki EM | 2Wiki F1 | 2Wiki Tokens | Hotpot EM | Hotpot F1 | Hotpot Tokens | Musique EM | Musique F1 | Musique Tokens |
>         | ReAct | 0.293 | 0.337 | 395,075 | 0.432 | 0.528 | 611,825 | 0.186 | 0.256 | 431,322 |
>         | Self-Ask | 0.083 | 0.148 | 469,714 | 0.31 | 0.391 | **532,756** | 0.147 | 0.2 | **292,195** |
>         | T2RAG | **0.693** | **0.775** | **366,199** | **0.623** | **0.732** | 606,914 | **0.391** | **0.491** | 336,555 |
>
>         | GPT-4o-mini |  |  |  |  |  |  |  |  |  |
>         | --- | --- | --- | --- | --- | --- | --- | --- | --- | --- |
>         | Method | 2Wiki EM | 2Wiki F1 | 2Wiki Tokens | Hotpot EM | Hotpot F1 | Hotpot Tokens | Musique EM | Musique F1 | Musique Tokens |
>         | ReAct | 0.534 | 0.598 | **448,156** | 0.586 | 0.729 | 697,667 | 0.325 | **0.458** | 369,656 |
>         | Self-Ask | 0.274 | 0.332 | 504,981 | 0.454 | 0.596 | **324,774** | 0.293 | 0.397 | 459,596 |
>         | T2RAG | **0.667** | **0.744** | 468,089 | 0.542 | 0.673 | 469,842 | **0.343** | 0.456 | **338,129** |
>
>         **Analysis**: T2RAG significantly outperforms both ReAct and Self-Ask across datasets, particularly in efficiency (tokens). While ReAct shows competitive performance on MuSiQue with GPT-4o-mini, T2RAG achieves comparable results with fewer tokens, highlighting its efficiency in complex multi-hop scenarios.
>
>
> Thanks again for the detailed and constructive suggestions! We have fixed typos, clarify some concerns, add more experiments per your suggestions
>
> ## References
>
> 1. From RAG to Memory:
> Non-Parametric Continual Learning for Large Language Models. In ICML 2025
> 2. HippoRAG:
> Neurobiologically Inspired Long-Term Memory for Large Language Models. In NeurIPS 2024
> 3. MiniRAG: Towards Extremely Simple Retrieval-Augmented Generation. Arxiv 2025
> 4. Interleaving Retrieval with Chain-of-Thought Reasoning
> for Knowledge-Intensive Multi-Step Questions. In ACL 2023
> 5. In-depth Analysis of Graph-based RAG in a Unified Framework, ArXiv 2025
> 6. Measuring and Narrowing the Compositionality Gap in Language Models. In EMNLP 2023
> 7. React: Synergizing reasoning and acting in language models. In ICLR 2022

---

### Official Review · Reviewer_jjQY · 2025-10-31

**Soundness:** 3
**Presentation:** 2
**Contribution:** 2
**Rating:** 4
**Confidence:** 4

**Summary:**

The paper proposes T2RAG, a novel framework for Retrieval-Augmented Generation (RAG) that improves multi-hop question answering by leveraging atomic triplets as the fundamental unit of retrieval and reasoning. Unlike traditional chunk-based or graph-based methods, T2RAG avoids costly graph construction and reduces token overhead by focusing on triplets with placeholders, which are resolved iteratively through an adaptive retrieval process. The paper demonstrates that T2RAG outperforms state-of-the-art RAG systems, achieving an 11% performance gain on various QA datasets while reducing retrieval costs by up to 45%.

**Strengths:**

1.	The paper proposes T2RAG, an innovative framework that leverages atomic knowledge triplets to go beyond traditional chunk- or graph-based retrieval-augmented generation (RAG) approaches.

2.	The authors demonstrate that T2RAG yields strong empirical gains across multiple QA datasets, improving average performance by 11% while reducing retrieval cost by up to 45%, which shows both effectiveness and efficiency.

**Weaknesses:**

1.	The related-work section could be strengthened. Although GraphRAG and multi-round RAG methods are discussed, the paper lacks comparison with other triplet-based retrieval methods such as SubgraphRAG, which also uses triplets for retrieval.

2.	Experimental parameter choices are not always fair. When comparing multi-round methods to direct chunk-retrieval baselines there is an information imbalance: some settings allow more total retrieved content than others.

3.	Several implementation details are unclear and need to be spelled out — for example, the paper does not explicitly state how the system behaves when there are no searchable triplets available.

**Questions:**

1.	The related-work section would benefit from a broader discussion of other triplet-based retrieval approaches. In particular, SubgraphRAG (Li et al., 2025) also employs triplet structures for retrieval and could serve as a valuable point of comparison. Including a discussion of such methods, as well as adding SubgraphRAG as an experimental baseline for both accuracy and retrieval efficiency, would help position T2RAG’s contribution more clearly within the current RAG landscape.

**Ref:** Simple is Effective: The Roles of Graphs and Large Language Models in Knowledge-Graph-Based Retrieval-Augmented Generation

2.	Although the paper highlights avoiding graph-construction overhead, it still depends on LLMs to extract triplets, which is conceptually similar to GraphRAG’s reliance on LLMs. Could the authors provide a more detailed analysis of the triplet-extraction step? In particular, do different extraction prompts significantly affect performance?
3.	How does the system handle queries whose triplets are all fuzzy (i.e., contain two or more placeholders)? If there are no searchable triplets, retrieval may fail — is there a fallback or mitigation strategy for this case?
4.	Regarding the fairness of comparisons: multi-round settings use N = 3, k = 5, which can retrieve up to 15 chunks for answering, while the direct chunk-retrieval baseline sets k = 5. This creates unequal information budgets. The authors should consider reporting the average number of chunks T2RAG actually uses and using that as a cap for other methods to ensure parity.
5.	When reporting token and time overheads for T2RAG, does the measurement include the cost of repairing or refining triplets, or does it only account for the final answer-generation phase? Please clarify.
6.	Typo: A space is missing in line 145 after "into multi-round".

If the authors can appropriately address the above issues, I would consider raising my score.

---

> ### Author Response · Authors · 2025-11-24
>
> We sincerely thank the reviewer for their thorough review, constructive feedback, and insightful questions, particularly regarding baseline comparisons and the nuances of our experimental setup. We have addressed each point below and incorporated these clarifications and additional results into the revised manuscript.
>
> ---
>
> > W1 & Q2: The related-work section could be strengthened. Although GraphRAG and multi-round RAG methods are discussed, the paper lacks comparison with other triplet-based retrieval methods such as SubgraphRAG,
> >
>
> We thank the reviewer for recommending SubgraphRAG. We agree that a comprehensive related work section is vital and have incorporated SubgraphRAG into our discussion on KGQA in Section 3 Related work - Graph RAG.
>
> - **Distinction from SubgraphRAG:**
> We clarify that SubgraphRAG primarily targets Knowledge Graph Question Answering (KGQA), which operates on pre-constructed, relatively clean Knowledge Graphs (like Wikidata). In contrast, our work focuses on **Open Domain QA (ODQA)**, where no such structured graph exists initially, presenting substantially different challenges regarding noise and structure. Additionally, SubgraphRAG focuses heavily on retrieval recall and often involves training (Retriever and GNN training), whereas T2RAG focuses on final answering performance in a training-free setting, leveraging the reasoning capabilities of modern LLMs.
> - **New Multi-Round Baselines (ReAct & Self-Ask):**
> To strengthen our ODQA baseline comparisons as requested, we have implemented and evaluated two additional multi-round methods adapted for our setting: **ReAct** [7] and **Self-Ask** [6].
>     - **Experimental Settings:**
>     • $k=5$, Max Iterations $=3$.
>     • **ReAct:** Uses `Thought`, `Action` (Search), and `Observation` loop.
>     • **Self-Ask:** Decomposes complex queries into simpler follow-up questions.
>
>         We report the EM, F1, Totoal tokens = input tokens + 4 * output tokens as follows.
>
>         **Results:**
>
>         | Gemini-2.5 flash |  |  |  |  |  |  |  |  |  |
>         | --- | --- | --- | --- | --- | --- | --- | --- | --- | --- |
>         | Method | 2Wiki EM | 2Wiki F1 | 2Wiki Tokens | Hotpot EM | Hotpot F1 | Hotpot Tokens | Musique EM | Musique F1 | Musique Tokens |
>         | ReAct | 0.293 | 0.337 | 395,075 | 0.432 | 0.528 | 611,825 | 0.186 | 0.256 | 431,322 |
>         | Self-Ask | 0.083 | 0.148 | 469,714 | 0.31 | 0.391 | **532,756** | 0.147 | 0.2 | **292,195** |
>         | T2RAG | **0.693** | **0.775** | **366,199** | **0.623** | **0.732** | 606,914 | **0.391** | **0.491** | 336,555 |
>
>         | GPT-4o-mini |  |  |  |  |  |  |  |  |  |
>         | --- | --- | --- | --- | --- | --- | --- | --- | --- | --- |
>         | Method | 2Wiki EM | 2Wiki F1 | 2Wiki Tokens | Hotpot EM | Hotpot F1 | Hotpot Tokens | Musique EM | Musique F1 | Musique Tokens |
>         | ReAct | 0.534 | 0.598 | **448,156** | 0.586 | 0.729 | 697,667 | 0.325 | **0.458** | 369,656 |
>         | Self-Ask | 0.274 | 0.332 | 504,981 | 0.454 | 0.596 | **324,774** | 0.293 | 0.397 | 459,596 |
>         | T2RAG | **0.667** | **0.744** | 468,089 | 0.542 | 0.673 | 469,842 | **0.343** | 0.456 | **338,129** |
>
>         **Analysis**: T2RAG significantly outperforms both ReAct and Self-Ask across datasets, particularly in efficiency (tokens). While ReAct shows competitive performance on MuSiQue with GPT-4o-mini, T2RAG achieves comparable results with fewer tokens, highlighting its efficiency in complex multi-hop scenarios.
>
>
> > W2 & Q4: Experimental parameter choices are not always fair. When comparing multi-round methods to direct chunk-retrieval baselines there is an information imbalance: some settings allow more total retrieved content than others.
> >
>
> We appreciate this comment and agree there may exist confusion.
>
> - First of all, we have a such fair comparison on **Figure 7**. We wish to clarify that Figure 7 (Performance on 2Wiki vs. top-k) already accounts for this "information imbalance." For multi-round methods, we calibrate the X-axis using Effective Top-K ($k \times \text{average number of iterations}$). This ensures we compare methods based on the total volume of information retrieved. The trend in Figure 7 demonstrates that T2RAG's performance remains consistently high and robust across different top-k values, confirming that its advantage stems from better reasoning and retrieval precision, not just access to more context. We will expand this analysis to include more datasets in the revised manuscript.
> - Regarding the main table, we follow the HippoRAG[2] paper, which also put the multi-round and single round method together in their Table 4. The reason is that the number of retrieved chunks can not be controlled in multi-round methods.

---

> ### Author Response · Authors · 2025-11-24
>
> > W3 & Q3: Several implementation details are unclear and need to be spelled out — for example, the paper does not explicitly state how the system behaves when there are no searchable triplets available.
> >
>
> Thank you for pointing out this edge case.
>
> - Currently, if no **newly searchable triplets** are generated and no remaining fuzzy triplets in an iteration, the loop terminates immediately, and the system proceeds to the final answer generation stage. We have clarified that in Section 4.
> - In practice, as valid questions typically contain at least one subject/object and predicate that form a searchable triplet, no searchable triplets happens very rare in the first step. If that happens in the query decomposition step or in the resolution step and some fuzzy triplets remain, our method will retrieve by the embedding of the query as a fallback and still keep the retrieval strategy (i.e., triplets as hook and number of chunks as budgets).
> - Your suggestion has inspired us to consider a "retry mechanism" where the LLM could re-attempt decomposition or resolution upon failure. While this could improve robustness, it introduces a trade-off between performance and efficiency. We have clarified our current handling of this edge case in the Section D.4.
>
> > Q2: Although the paper highlights avoiding graph-construction overhead, it still depends on LLMs to extract triplets, which is conceptually similar to GraphRAG’s reliance on LLMs. Could the authors provide a more detailed analysis of the triplet-extraction step? In particular, do different extraction prompts significantly affect performance?
> >
>
> We thank the reviewer for this question regarding triplet extraction.
>
> - Triplet extraction is indeed a foundational step for most ODQA GraphRAG methods [1,2,3,4,5] in the absence of a pre-existing KG. To ensure a fair comparison, we intentionally followed the most popular and standardized setting: using **GPT-4o-mini** with the same extraction prompts as **HippoRAG2 [1]**.
> - While not the primary focus of our novelty, we validated the quality of this step. As shown in **Table 5**, our manual evaluation of 50 samples yielded **>95% Recall and Precision**. This confirms that the extraction process is reliable enough to support our subsequent retrieval and reasoning stages.
>
> > Q5: When reporting token and time overheads for T2RAG, does the measurement include the cost of repairing or refining triplets, or does it only account for the final answer-generation phase?
> >
>
> We confirm that our reported measurements **include all costs**: query decomposition, triplet refinement, intermediate retrieval, and final answer generation. If "thinking" (Chain-of-Thought) is enabled, those tokens are also counted.
>
> We believe **Total Tokens** is superior efficiency metric compared to wall-clock time, as they are agnostic to network latency and hardware variations (CPU vs. GPU). We have included the details of token consumption for both the offline and online stage in Appendix B.
>
> We once again thank the reviewer for their valuable feedback, which has significantly helped us refine our work. We are committed to fixing typos, clarifying concerns, and adding the suggested experiments in the final revision.
>
> ## References
>
> 1. From RAG to Memory:
> Non-Parametric Continual Learning for Large Language Models. In ICML 2025
> 2. HippoRAG:
> Neurobiologically Inspired Long-Term Memory for Large Language Models. In NeurIPS 2024
> 3. LightRAG: Simple and Fast Retrieval-Augmented Generation. In EMNLP2025
> 4. From Local to Global: A Graph RAG Approach to Query-Focused
> Summarization. Arxiv 2024
> 5. MiniRAG: Towards Extremely Simple Retrieval-Augmented Generation. Arxiv 2025
> 6. Measuring and Narrowing the Compositionality Gap in Language Models. In EMNLP 2023
> 7. React: Synergizing reasoning and acting in language models. In ICLR 2022

---

### Official Review · Reviewer_ySAV · 2025-11-01

**Soundness:** 3
**Presentation:** 3
**Contribution:** 2
**Rating:** 4
**Confidence:** 4

**Summary:**

This paper proposes T2RAG, a framework to conduct GraphRAG without actually building knowledge graphs (KGs), which may help avoid the costly, time-consuming, and error-prone process of offline KG construction. This is done by extracting triplets from the corpus and then storing these triplets directly.
To store these triplets, T2RAG converts each triplet into natural language, embeds them as dense vectors, and stores them in vector databases. Given a question, the LLMs would be asked to decompose the query into triplets, and these triplets are then embedded and thus used to search relevant triplets in the triplet vector datasets to resolve the questions. Experiments show that T2RAG can improve some QA datasets significantly.

**Strengths:**

- The idea of using purely triplet-based databases for multi-hop reasoning with RAG is interesting.
- The proposed online retrieval method is also interesting. With those placeholders, the query triplets can be semantically closer to the relevant triplets in the vector database.
- For QA datasets that do not require that many hops, the proposed method may improve accuracy significantly.

**Weaknesses:**

- One reason to use graph data is that it can be more friendly when dealing with multi-hop reasoning. Building databases without entity links but with only triplets essentially gives up graph structures during offline building. So, during the online process, the method would, in principle, need to somehow reconstruct the multi-hop links.
    - This can be especially hard for long-hop reasoning, and this may also be the reason why T2RAG mostly improves datasets with fewer hops, e.g., PopQA, 2Wiki.
    - So, it would be more interesting to see whether T2RAG can still perform decently for reasoning with even longer hops. And in such cases, what the costs would be.
- In the meantime, even though offline building of KGs can have some issues, storing things in triplet embedding format would also cause issues, especially when dealing with super large corpora.
    - For example, the number of triplets can grow fast, and the embedding quality would then affect the retrieval quality more significantly. With methods like FAISS for efficiency, more noise would also be introduced.
    - Therefore, it should also be required to test T2RAG on super large corpora.

**Questions:**

See above.

---

> ### Author Response · Authors · 2025-11-24
>
> We sincerely thank the reviewers for their thorough feedback and insightful comments. We have addressed each point below and incorporated these clarifications and additional results into the revised manuscript in Appendix G.
>
> > W1：Building databases without entity links but with only triplets essentially gives up graph structures.  …The method would, in principle, need to somehow reconstruct the multi-hop links.
> >
>
> We appreciate the reviewer's concern regarding the potential loss of structural information when bypassing explicit offline graph construction. However, we respectfully argue that our approach does not "give up" graph structures, but rather shifts the construction from **offline indexing** to **online inference**.
>
> **1. Implicit Inference-Time Graph Construction**
> While recent works focus on explicit graph construction during inference [1, 2, 3], few explore **implicit graph construction**—providing the necessary nodes (entities) and edges (relations) at the final answering stage without pre-hardwired connections. T2RAG allows the LLM to construct the necessary graph structures internally within its context window. This offers distinct advantages:
>
> - **Handling Ambiguity:** It naturally handles entity ambiguity and diverse reasoning structures (chains, trees, or DAGs) dynamically based on the query, rather than relying on a rigid, pre-defined schema.
> - **Leveraging Reasoning Models:** Our experiments show that T2RAG significantly benefits from the enhanced reasoning capabilities of state-of-the-art models (e.g., Gemini-1.5-Pro, o1). In contrast, methods relying on rigid graph pre-processing (like HippoRAG or RAPTOR) may not fully capitalize on these reasoning gains, as the "hard" graph structure can sometimes constrain the "soft" reasoning power of the LLM.
>
> **2. The "Locality Assumption" and Retrieval Equivalence**
> We argue that complex offline graph construction is often unnecessary for effective retrieval due to the **locality assumption, especially for questions within 3 hops.**
>
> - First, In Knowledge Graphs, a text chunk typically forms a star graph or a densely connected cluster. Second, Traversing a pre-built subgraph (as done in GraphRAG) yields a similar information set to our method, which retrieves all triplets associated with a relevant chunk. T2RAG aggregates these triplets and allows the LLM to filter or combine them using the decomposed query. This achieves the same "PageRank-style" filtering effect but transfers the computational burden to the highly capable LLM during inference, rather than the indexing stage.
>
> **3. Efficiency and Performance Trade-off**
> Our primary contribution is maximizing efficiency compared to GraphRAG and performance compared to multi-round RAG. T2RAG offers a robust balance, particularly for 1-2 hop QA applications, while retaining the potential for deeper reasoning as base models improve.
>
> > W1.1: This can be especially hard for long-hop reasoning, and this may also be the reason why T2RAG mostly improves datasets with fewer hops…; So, it would be more interesting to see whether T2RAG can still perform decently for reasoning with even longer hops.
> >
>
> To address the concern regarding performance on longer reasoning chains, we evaluated T2RAG against strong baselines (HippoRAG2 and IRCoT) on the MuSiQue dataset, stratified by hop count (2, 3, and 4 hops).
>
> | Method | 2-hop F1 | 2-hop EM | 3-hop F1 | 3-hop EM | 4-hop F1 | 4-hop EM |
> | --- | --- | --- | --- | --- | --- | --- |
> | HippoRAG2 | 58.79 | 43.82 | 41.90 | 27.22 | **31.14** | **22.29** |
> | IRCoT | 43.12 | 33.20 | 31.81 | 22.78 | 13.22 | 7.83 |
> | T2RAG | **60.81** | **48.46** | **42.50** | **31.33** | 24.46 | 14.46 |
>
> **Analysis:** The results demonstrate that T2RAG consistently achieves State-of-the-Art performance on 2-hop and 3-hop queries. While graph traversal methods like HippoRAG2 show an advantage at extreme hop counts (4-hop) due to global graph connectivity, T2RAG still perform decently over standard iterative approaches like IRCoT across all complexity levels. As a result, we recommend using T2RAG for questions with no more than 3 hops.

---

> ### Author Response · Authors · 2025-11-24
>
> > W2: Even though offline building of KGs can have some issues, storing things in triplet embedding format would also cause issues, especially when dealing with super large corpora. …  the embedding quality would then affect the retrieval quality. Methods like FAISS for efficiency, more noise would also be introduced.
> >
>
> **1. Embedding Quality and Noise**
> The reviewer raised concerns that triplet embeddings might introduce noise or quality issues compared to graph structures. We argue that triplets actually **reduce** noise compared to chunk-level embeddings.
>
> - **Granularity:** A chunk embedding averages multiple facts into a single vector, often diluting specific details. Triplet embeddings isolate specific relationships (`Subject-Predicate-Object`), resulting in cleaner, more precise retrieval signals.
> - **Modern Encoders:** With the rapid development of LLM-based embedding models, the semantic quality of triplet representations is high, and the dimensionality is manageable. The quality can be further improved with larger number of dimensions or model parameters [5].
>
> **2. FAISS Efficiency**
> Regarding the concern about FAISS and approximation errors:
>
> - The approximation error in modern ANN (Approximate Nearest Neighbor) libraries like FAISS is theoretically bounded.
> - We have validated our method on datasets ranging from small corpora (HotpotQA, ~914k tokens) to larger ones (PopQA, ~2.7M tokens). These are standard, widely-accepted benchmarks in the field [4].
> - We note that noise from embedding retrieval is a universal challenge that affects all dense retrieval methods (including those used to construct graphs or retrieve chunks in baselines), not just T2RAG.
>
> We thank the reviewer again for all questions and suggestions. The above revision has been reflected in the revised version in Appendix G.
>
> ## References
>
> 1. HippoRAG:
> Neurobiologically Inspired Long-Term Memory for Large Language Models. In NeurIPS 2024
> 2. Think-on-graph: Deep and responsible reasoning of large language model on knowledge graph. In ICLR 2024
> 3. Reasoning on graphs: Faithful and Interpretable Large Language Model Reasoning. In ICLR 2024
> 4. In-depth Analysis of Graph-based RAG in a Unified Framework, ArXiv 2025
> 5. MTEB: Massive Text Embedding Benchmark, In EACL 2023

---

### Note · Authors · 2026-01-03

I have read and agree with the venue's withdrawal policy on behalf of myself and my co-authors.